# Re-FORC: Adaptive Reward Prediction for Efficient Chain-of-Thought Reasoning

**Renos Zabounidis** [1 2]  **Aditya Golatkar** [1]  **Michael Kleinman** [1]  **Alessandro Achille** [1]  **Wei Xia** [1]  **Stefano Soatto** [1]

## Abstract

We propose Re-FORC, an adaptive reward prediction method that, given a query, enables prediction of the expected future rewards as a function of the number of future thinking tokens. Re-FORC trains a lightweight adapter on reasoning models, demonstrating improved prediction with longer reasoning and larger models. Re-FORC enables: 1) *early stopping* of unpromising reasoning chains, reducing compute by up to 24% compared to fixed-budget cutoffs, while maintaining accuracy, 2) *optimized model and thinking length selection* that outperforms the largest model alone—reaching 1.7 percentage points higher peak accuracy while needing up to 12% less compute to match the largest model's accuracy, 3) *adaptive test-time scaling*, which increases accuracy by 8.8 percentage points (on average at maximum compute) over confidence-based baselines. Re-FORC allows dynamic reasoning with length control via cost-per-token thresholds while estimating computation time upfront.

## 1. Introduction

Modern reasoning models can dynamically use inference-time computation to improve answer quality, but determining the optimal amount of inference-time computation for a given query remains an open challenge. While an inference pipeline can generate longer reasoning traces, backtrack to explore alternatives, or even delegate computation to specialized models, the large action space makes it difficult to identify the best strategy for any given query. This challenge is compounded by the diversity of user requirements. Different users have varying tolerance for latency and assign different values to output quality. A strategy that works well

for a time-sensitive application may be entirely inappropriate for a high-stakes decision where accuracy is paramount. How can we adaptively optimize inference-time compute based on both query difficulty and user-specific constraints?

The question can be framed more generally as maximizing the net utility of inference:

$$J = \mathbb{E}[R_{best} - \lambda T_{\text{total}}]$$

where $R_{best}$ represents the reward of the best answer found during inference-time reasoning (i.e., variable-length chain-of-thought computation), $T_{\text{total}}$ is the total compute cost incurred, $\lambda$ captures the user's cost sensitivity, and the expectation is over the randomness in inference.

While models can be trained for specific trade-offs, dynamically adapting to user constraints at inference time remains largely unexplored. Crucially, there is no canonical choice of $\lambda$ when evaluating $J$, since the value of an agent depends on the user and the environment (Achille & Soatto, 2026). Consider an agent tasked with determining whether a fruit is edible: if the agent takes so long that the fruit has spoiled, providing the correct answer is of no use. But if the agent is a botanist categorizing species for an encyclopedia, it matters little whether the fruit is edible by the time the answer arrives. In general, an agent must learn to modulate the cost of time depending on the environment and the user's stated preferences.

Our key insight is that current models lack the **ability to predict the marginal benefit of additional computation** for a given query. Lacking this, we cannot make informed decisions about *when to continue reasoning, when to backtrack, or when the expected improvement no longer justifies the computational cost.*

To address this gap, we introduce Re-FORC, a method for forecasting the reward-versus-compute trade-off both before and during inference in reasoning-based LLMs. Given a partial reasoning trace, Re-FORC predicts the distribution of expected rewards from generating additional thinking tokens. This capability is achieved through a lightweight adapter fine-tuned on top of existing models, enabling predictions for both the base model and external black-box

Work done during an internship at AWS Agentic AI. [1]AWS Agentic AI [2]Carnegie Mellon University. Correspondence to: Aditya Golatkar <agolatka@amazon.com>.

*Proceedings of the $43^{rd}$ International Conference on Machine Learning*, Seoul, South Korea. PMLR 306, 2026. Copyright 2026 by the author(s).

systems.

Building on Re-FORC, we then introduce a **greedy algorithm for optimal decision-making** inspired by the theory of Pandora's box problems (Weitzman, 1979). This development is motivated by the following practical application requirements:

- **Adaptive early termination:** By stopping reasoning trajectories when the marginal benefit of additional computation no longer outweighs its cost, Re-FORC reduces computational cost by up to 24% on average compared to fixed-budget cutoffs (S1, Muennighoff et al., 2025) while maintaining accuracy.
- **Joint model and compute optimization:** When multiple model sizes are available, Re-FORC jointly selects both the optimal model and the number of reasoning tokens to maximize net utility for each query, going beyond traditional model routing by explicitly considering compute budgets.
- **User-controlled inference:** Users can specify their cost-performance trade-off $\lambda$ at inference time without model retraining, providing a more intuitive alternative to token-count specifications while adapting inference to query complexity.
- **Transparent compute estimation:** Re-FORC can provide users with upfront estimates of expected computation time, improving the user experience for latency-sensitive applications.

## 2. Related Work

Our work intersects two key research areas: forecasting methods that predict LLM performance, and efficient reasoning strategies that optimize compute-accuracy trade-offs during inference.

**Forecasting, Probing, and Verifier-Guided Inference-Time Scaling.** Most forecasting research addresses non-chain-of-thought contexts, predicting restart benefits (Manvi et al., 2024) or initial correctness (Damani et al., 2025). Recent work shows models encode future correctness (Zhang et al., 2025; Yoon et al., 2025) and factuality signals (Servedio et al., 2025; Yang & Jia, 2025) during reasoning. While some approaches use external verifiers for intermediate step evaluation (Uscidda et al., 2025; Snell et al., 2025; Wu et al., 2025), our method forecasts future expected reward as a continuous function of additional reasoning tokens, enabling utility-based decisions across reasoning horizons. Concurrent work on adaptive test-time compute allocation shows that compute-optimal scaling policies (Snell et al., 2025) and reward-guided adaptive reasoning depth (Cui et al., 2025) can outperform naive scaling approaches. Tree-of-thought methods enable deliberate multi-branch search over reasoning states (Long, 2023; Besta et al., 2025) and adaptive

budget allocation across multiple trajectories (Liao et al., 2025), though recent work identifies efficiency challenges in naive verifier-guided exploration (Wang et al., 2025). Recent iterative refinement methods repeatedly generate and aggregate reasoning traces, showing near-monotonic accuracy gains but rapidly growing compute cost (Venkatraman et al., 2025). Our forecaster differs in that it predicts a continuous marginal value curve of extra thinking tokens, rather than heuristically expanding and pruning reasoning paths.

**Efficient Reasoning and Model Selection.** A parallel line of work has developed strategies for efficient reasoning under compute constraints. Self-consistency (Wang et al., 2023b) spawned compute-aware variants that halt when votes converge (Li et al., 2024; Liu & Wang, 2025) or use confidence-weighted aggregation (Taubenfeld et al., 2025). Theoretical analyses characterize diminishing marginal returns from additional samples (Komiyama et al., 2026). Recent work on adaptive reasoning length control includes token-budget-aware policies (Yuan et al., 2025) and learned stopping mechanisms that predict when further reasoning becomes redundant (Sun et al., 2025), including dynamic early-exit policies that truncate reasoning once additional steps yield diminishing returns (Yang et al., 2025b). Structured exploration increases search breadth under larger budgets (Bi et al., 2025). At the token level, early exit mechanisms enable anytime generation with calibrated stopping (Schuster et al., 2022), while agentic frameworks learn when to plan (Paglieri et al., 2025; Pan & Zhao, 2025) and strategic decomposition approaches like ReAct (Yao et al., 2023), SCALAR (Zabounidis et al., 2025), least-to-most prompting (Zhou et al., 2023), and plan-and-solve strategies (Wang et al., 2023a) allocate deliberate planning compute. Separately, model routing systems choose among different-sized models to optimize accuracy-cost trade-offs (Jitkrittum et al., 2026; Guha et al., 2024; Ding et al., 2025; Yue et al., 2025; Guo et al., 2025). In addition to model selection, recent works aim to control the reasoning length either through training-free (Muennighoff et al., 2025) or training-based approaches (Aggarwal & Welleck, 2025; Kleinman et al., 2025). A complementary line of work uses model-internal confidence signals—such as token-level entropy or distributional confidence—to halt or aggregate reasoning traces, with DeepConf (Fu et al., 2026) being a recent representative; we compare against DeepConf as a strong adaptive baseline. Our approach bridges these areas by using forecasted reward curves to make principled decisions about both when to stop reasoning and which model to use, grounded in metareasoning theory (Russell & Wefald, 1991; Weitzman, 1979; Aouad et al., 2025). Recent work frames modern LLM inference as bounded-optimal decision-making over cognitive effort (Fu et al., 2026), providing theoretical grounding for treating compute allocation as an economically principled optimization problem. Addi-

tionally, Achille & Soatto (2026) recently framed inference-time search as a Pandora's box problem and proposed using forecasted rewards for making cost-aware decisions, which we realize in our work. In concurrent work, Manvi et al. (2026) also train a forecaster and apply it for efficient and adaptive reasoning.

## 3. Methodology

Inference-time compute allocation in reasoning models presents a sequential decision problem where an agent must decide at each step whether to generate additional thinking tokens or terminate. Let $\mathcal{X}$ denote the query space, $\mathcal{Z}$ the space of partial reasoning traces (possibly multiple trajectories), and $\mathcal{Y}$ the output space, and $\Pi = \{\pi_i\}_{i=1}^N$ be a collection of reasoning models. At each time step, the system state is $(x, z)$ where $x \in \mathcal{X}$ is the query, $z \in \mathcal{Z}$ is the collection of partial traces generated so far, and the agent needs to decide which trajectory to continue or terminate the search.

This problem exhibits the structure of a *Markov chain selection* problem (Scully & Terenin, 2025), where the agent must choose which of multiple stochastic processes (reasoning trajectories for LLMs) to advance. Each reasoning continuation corresponds to a transient Markov chain with reward structure, and the agent must select which chain to progress based on expected net utility. However, unlike classical settings where reward distributions are known, reasoning models operate with unknown, state-dependent reward distributions that depend on query complexity and current reasoning progress. Recently, (Achille & Soatto, 2026) showed that universal search (Levin, 1973) can be formulated as a Pandora's box problem which can be solved with the Gittins policy. We use the framework from (Achille & Soatto, 2026) in Section 4.2 where we propose the Pandora's box greedy search for reasoning models using our forecaster $\psi(t \mid x, z, \pi)$.

The optimal policy for such problems (Weitzman, 1979) takes the form of a Gittins index policy (Scully & Terenin, 2025; Xie et al., 2024), which assigns each possible continuation a reservation value—the minimum expected reward improvement needed to justify its computational cost. In the Gittins index policy (Scully & Terenin, 2025), we compute the Gittins index of each continuation, and compare it against our estimate of current best reward. We terminate search if none of the continuations improve the current best reward, otherwise we choose the trajectory with highest Gittins index.

The central challenge is that computing Gittins indices requires knowledge of the reward distributions for different reasoning continuations, which are unknown and must be learned. We address this by introducing **Re-FORC**, which learns to predict the forecasting functional $\psi(t \mid x, z, \pi)$ that estimates expected rewards from generating $t$ additional thinking tokens from state $(x, z)$ using model $\pi$. Using our forecaster we can approximate the Gittins index to choose the next action in the Markov chain.

We now introduce the necessary preliminaries and define the forecasting functional that enables training our predictor to approximate the Gittins index.

### 3.1. Sequential Compute Allocation

We formalize the inference-time compute allocation problem in LLMs by defining the decision space, objectives, and constraints. The core challenge is modeling how reasoning models generate thinking tokens and how the quality of their final answers depends on the computational resources allocated.

We use a Markov decision process where the state space is $\mathcal{S} = \mathcal{X} \times \mathcal{Z}$ and each state $s = (x, z)$ represents a query with a partial reasoning trace. The action space $\mathcal{A}$ includes:

- **Continue reasoning**: Generate $\Delta$ additional thinking tokens
- **Terminate**: Stop reasoning and output final answer $y$
- **Switch model**: Transfer to a different reasoning model $\pi'$ when available

Each reasoning continuation from state $(x, z)$ can be modeled as advancing a transient Markov chain with reward function $R : \mathcal{X} \times \mathcal{Y} \to [0, 1]$ and cost function $c : \mathcal{A} \to \mathbb{R}_+$. The agent's objective is to maximize expected net utility:

$$J = \mathbb{E}[R_{best} - \lambda \cdot T_{\text{total}}] \tag{1}$$

where $R_{best}$ is the reward of the best answer discovered, $T_{\text{total}}$ is the total computational cost, and $\lambda > 0$ represents the cost sensitivity parameter. This objective balances exploration of potentially better solutions against computational expenditure measured in (necessarily subjective, environment- and user-dependent) units of $\lambda$.

The key challenge distinguishing our setting from classical Markov chain selection problems lies in computing the Gittins index itself. For a Markov chain in state $s$, the Gittins index $g$ for a reasoning trajectory is defined as the solution to:

$$\mathbb{E}\left[(R(x, y) - g)_+ \Big| s = (x, z)\right] - \lambda t = 0 \tag{2}$$

where $t$ is the number of future reasoning tokens and $(x)_+ = \max(x, 0)$. For binary rewards $R \in \{0, 1\}$ with success probability $p$, the Gittins index has closed form:

$$g = 1 - \frac{\lambda t}{p} \tag{3}$$

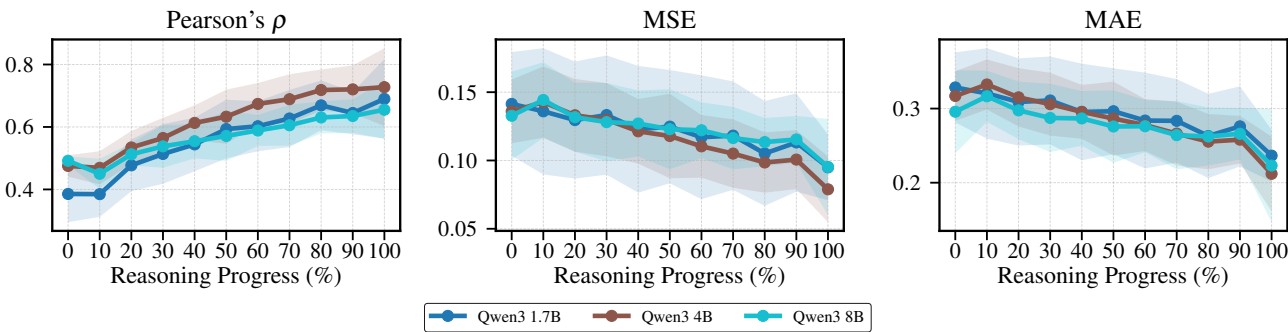

*Figure 1.* **Forecast performance with reasoning progress.** Correlation between Re-FORC (Equation (7)) and the true reward (Equation (6)) as the chain-of-thought progresses for Qwen3 (Yang et al., 2025a) models, averaged across four math benchmarks: MATH500, AMC 2024, AIME 2024, and AIME 2025. **(left)** Pearson correlation $\rho$ (higher is better); **(middle)** mean squared error (MSE, lower is better); **(right)** mean absolute error (MAE, lower is better). Forecast quality improves with reasoning progress. Per-dataset results are in Appendix B.1.

when $p > \lambda t$ (positive expected value). Given our forecaster $\psi(t \mid x, z, \pi)$ which predicts $p$ for $t$-token continuations, we define the optimal compute budget as:

$$t^*(x, z, \pi) = \arg\min_t \left[ \frac{\lambda t}{\psi(t \mid x, z, \pi)} \right] \quad (4)$$

The Gittins index for model $\pi_i$ at state $(x, z)$ is then:

$$g_i(x, z) = 1 - \frac{\lambda_i t^*(x, z, \pi_i)}{\psi(t^*(x, z, \pi_i) \mid x, z, \pi_i)} \quad (5)$$

The Gittins index policy selects the option with highest $g_i$ and terminates when all $g_i < R_{best}$, where $R_{best}$ is the best reward obtained so far.

Generally, computing the Gittins index requires solving Equation (2), which depends on the reward distribution. When the outcome is binary, this reduces to calculating the expected reward $\mathbb{E}[R(x, y)]$. This expectation depends on the stochastic reasoning process and the final answer quality, neither of which have closed-form expressions for language models. We address this challenge by learning the forecasting functional $\psi(t \mid x, z, \pi) = \mathbb{E}[R(x, y)]$ for $t$-token continuations. This functional enables approximate computation of Gittins indices and principled decision-making in our adaptive setting.

### 3.2. Adaptive Reward Prediction

Given a query $x \in \mathcal{X}$, a partial chain-of-thought $z \in \mathcal{Z}$, and a reasoning model $\pi$, we define two modes of inference: $\pi^{(r)}$ for thinking token generation and $\pi^{(o)}$ for final output generation. The output given $t$ additional thinking tokens is obtained by first sampling additional reasoning tokens $z_t \sim \pi^{(r)}(\cdot | x, z, t)$ where $|z_t| \leq t$, then sampling the output $y \sim \pi^{(o)}(\cdot | x, z, z_t)$.

Given a reward function $R(x, y) : \mathcal{X} \times \mathcal{Y} \to [0, 1]$, the adaptive forecasting functional is:

$$\psi(t \mid x, z, \pi) \triangleq \mathbb{E}_{\substack{z_t \sim \pi^{(r)}(\cdot|x,z,t) \\ y \sim \pi^{(o)}(\cdot|x,z,z_t)}} [R(x, y)] \quad (6)$$

This functional represents the expected reward after allocating exactly $t$ additional thinking tokens, starting from the current reasoning state $(x, z)$. Critically, $\psi$ is query-dependent, path-dependent, and accounts for the stochastic nature of both reasoning generation and final answer sampling.

To predict $\psi(t \mid x, z, \pi)$, we design a lightweight forecasting module that can be attached to existing reasoning models. We model the forecaster output using a $\mathrm{Beta}(\alpha, \beta)$ distribution: it has bounded support on $[0, 1]$ matching the reward range, captures a wide range of confidence patterns with two parameters, and yields both a mean prediction and a variance estimate for calibrated uncertainty. Numerically, applying softplus to the network outputs keeps $\alpha, \beta > 0$ during training and the log-likelihood is straightforward to optimize.

The forecaster predicts Beta parameters $[\alpha_\theta(x, z, t), \beta_\theta(x, z, t)]^T$ for each thinking token budget $t$. At inference, we use the Beta mean as our point estimate of expected reward:

$$\hat{\psi}(t \mid x, z, \pi) = \frac{\alpha_\theta(x, z, t)}{\alpha_\theta(x, z, t) + \beta_\theta(x, z, t)} \quad (7)$$

Since $\psi$ is defined over discrete token budgets $t \in \mathbb{N}$, we predict it on a uniform grid $\mathcal{T} = \{0, \Delta, 2\Delta, \ldots, t_{max}\}$ and obtain values at arbitrary $t$ by linear interpolation between adjacent grid points. This approach balances computational efficiency with forecasting accuracy.

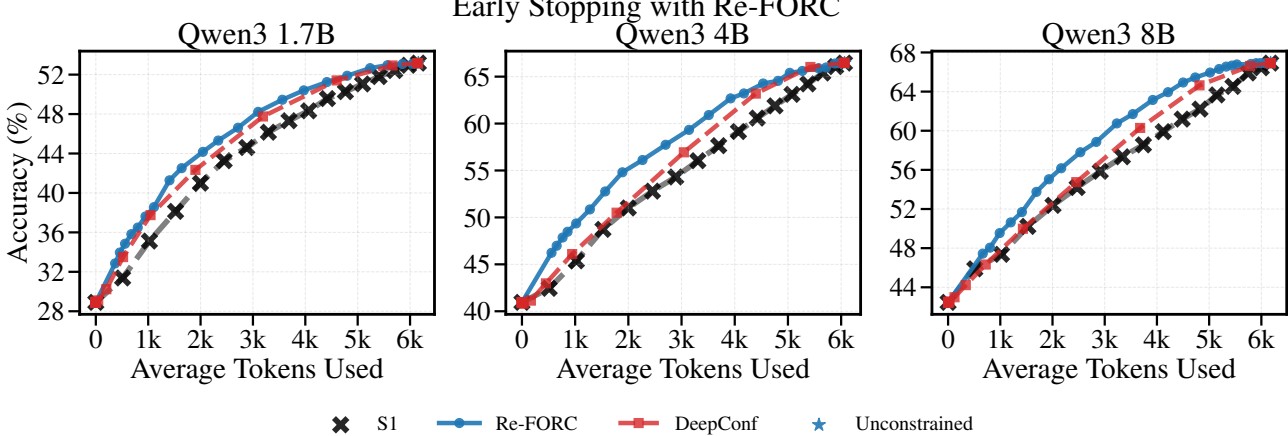

*Figure 2.* **Early stopping with Re-FORC.** Accuracy vs. average tokens used for Qwen3 1.7B **(left)**, 4B **(middle)**, and 8B **(right)**, averaged across four math benchmarks: MATH500, AMC 2024, AIME 2024, and AIME 2025 (see Section 5.2). Re-FORC (Equation (11)) is compared against S1 (Muennighoff et al., 2025), the recent confidence-based DeepConf baseline (Fu et al., 2026), and unconstrained generation. For the DeepConf baseline, we stop the trajectory early if the confidence of the preceding 512 tokens is below a threshold, and evaluate across thresholds. Re-FORC dominates all baselines at nearly every token budget across all three model sizes, with the gap largest in the moderate-compute regime. The cost–accuracy trade-off is controlled by a single parameter $\lambda$. Per-dataset result are in Appendix B.2

### 3.3. Training the predictor

Training the adaptive reward forecaster requires generating a dataset of $(x, z, t, r)$ tuples where $r$ represents the true expected reward $\psi(t \mid x, z, \pi)$ from continuing reasoning for $t$ additional tokens from state $(x, z)$. We generate training data by sampling problem instances $(x_i, y_i)$ and generating full unconstrained reasoning trajectories up to the maximum context length. From each complete trajectory, we extract partial traces $z$ by truncating at regular token intervals corresponding to our forecasting grid $\mathcal{T} = \{0, \Delta, 2\Delta, \ldots, t_{\max}\}$. For each partial trace $z$ truncated at position $\ell$, we sample the model's answer directly from state $(x, z)$.

When constructing the adaptive forecasting functional $\psi(t \mid x, z, \pi)$ for different continuation lengths $t$, we reuse these sampled answers by taking all trajectory segments that extend exactly $t$ tokens beyond the truncation point $\ell$. This provides an efficient Monte Carlo approximation through trajectory reuse compared to the alternative of generating multiple continuations for every partial trajectory and every forecasting horizon would require $O(|\mathcal{T}| \times N \times L)$ trajectory samples, where $N$ is the number of Monte Carlo samples per estimate and $L$ is the maximum trajectory length. Our reuse strategy reduces this to $O(N)$ samples total while maintaining unbiased estimates of $\psi(t \mid x, z, \pi)$ across all forecasting horizons.

The forecaster is trained by maximizing the likelihood of observed rewards under the predicted Beta distributions:

$$\mathcal{L}_{\text{forecast}} = \mathbb{E}_{(x,z,t,r) \sim D_{\text{forecast}}} \big[ -\log \text{Beta}(\alpha_\theta(x, z, t),$$
$$\beta_\theta(x, z, t))(r) \big] \quad (8)$$

We provide more training details in Section 5. Note that our forecasters are lightweight adapters attached to frozen base reasoning models.

## 4. Compute-Aware Inference

In this section, we explore the applications of our forecaster $\psi(t \mid x, z, \pi)$ for optimal inference-time decision-making. We apply the Gittins index-inspired greedy algorithm (Scully & Terenin, 2025; Xie et al., 2024) to evaluate the expected improvement from continuing each available reasoning trace and select the action with highest expected net utility at each step in the search. This approach enables four key applications: (1) early termination decisions that halt partially-completed reasoning traces when marginal improvement no longer justifies compute costs, (2) initial model selection that chooses the optimal model for query processing based on expected cost-accuracy trade-offs, (3) dynamic reselection that transfers unsuccessful queries to models with potentially better capabilities, and (4) exploration-exploitation trade-offs that balance sampling new reasoning traces against continuing existing ones.

### 4.1. Early stopping

Let $z^*$ be the best (partial) thinking trace, with reward $R_{best}$. Let $z$ be the current thinking trace from a model $\pi_i$ (potentially $z = z^*$). If the agent decides to extend $z$ for additional $t$ steps, the expected improvement in reward is:

$$\Delta_t J = \mathbb{E}_{\substack{y \sim \pi^{(o)}(\cdot \mid x, z, z_t) \\ z_t \sim \pi^{(r)}(\cdot \mid x, z, t)}} \big[ (R(x, y) - R_{best})_+ \big] - \lambda t \quad (9)$$

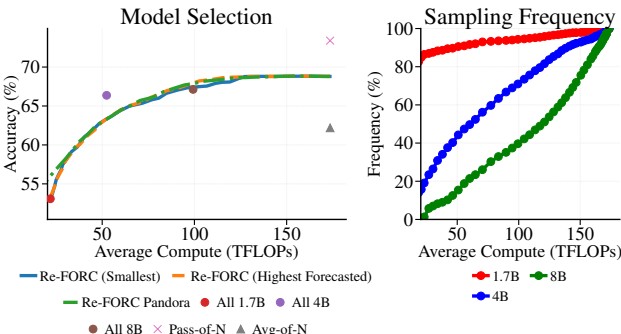

*Figure 3.* **Model and thinking-length selection with Re-FORC. (Left)** Accuracy vs. average compute per problem (TFLOPs) for Qwen3 models averaged across four math benchmarks (MATH500, AMC 2024, AIME 2024, AIME 2025), using the model-selection strategies in Equation (13). The three Re-FORC variants are: *Re-FORC (Smallest)*, which orders models from cheapest to most expensive; *Re-FORC (Highest Forecasted)*, which orders models by decreasing Gittins index; and *Re-FORC Pandora*, which interleaves reasoning across models at the step level (Equation (12)). All Re-FORC variants outperform individual-model baselines in peak accuracy, reaching 1.7 percentage points above All-8B while needing 10% less compute on average to match its accuracy. Pass-of-N is an oracle upper bound. **(Right)** Sampling frequency of Re-FORC Pandora across model sizes as a function of compute: at minimum compute it preferentially samples the 1.7B model, while at maximum compute it draws from all three. Per-dataset breakdowns in Appendix B.3.

where $(\cdot)^+$ denotes the positive part—even if the new reward is lower than $R_{best}$, we can discard that attempt, but we still pay the compute cost $\lambda t$.

For a binary reward distribution, the expected improvement simplifies to:

$$\Delta_t J = \psi(t \mid x, z, \pi)(1 - R_{best}) - \lambda t \qquad (10)$$

We continue reasoning if $\Delta_t J > 0$ for some $t^*$, which is equivalent to continuing iff the Gittins index from Equation (5) exceeds the current best reward:

$$g_i(x, z) = 1 - \frac{\lambda t^*}{\psi(t^* \mid x, z, \pi)} > R_{best} \qquad (11)$$

Otherwise, we stop reasoning. When $\lambda$ is high or $R_{best}$ is close to 1, the condition is harder to satisfy, leading to earlier termination. In practice we use our forecaster (Equation (7)) to estimate $\psi$.

In Figure 2 we show that early stopping with Re-FORC significantly improves the reward compute trade-off across different sizes of reasoning models averaged across 4 math datasets (Section 5.2). For instance, compared to the S1 baseline (Muennighoff et al., 2025), we save 24% compute on average for the 4B reasoning model while preserving accuracy.

## 4.2. Pandora's Box Greedy Search

We extend early stopping to the multi-model setting using the Gittins index policy (Achille & Soatto, 2026). Given a collection $\{\pi_1, \ldots, \pi_k\}$ of reasoning models with per-token costs $\lambda_i$, and partial traces $\{z_1, \ldots, z_n\}$, we compute the Gittins index $g_i(x, z_j)$ for each model-trace pair using Equation (5). At each step, we select the pair with highest Gittins index:

$$(i^*, j^*) = \arg\max_{i,j} g_i(x, z_j) \qquad (12)$$

and continue trace $z_{j^*}$ using model $\pi_{i^*}$ for $\Delta$ tokens (1 timestep). We add the new trace to the set of explored trajectories and update $R_{best}$ if needed. We terminate when all $g_i(x, z_j) < R_{best}$, where $R_{best}$ is the best reward so far. In practice, we use $\hat{\psi}(t|x, z, \pi)$ to approximate the forecasting functional. With a single model and trace, this reduces to Re-FORC-stopping (Equation (11)).

## 4.3. Model Selection

Model selection is a special case of the Pandora's box greedy search (Equation (12)) restricted to at most one trajectory per model. The Gittins index ranks models by expected net utility before reasoning begins:

$$j = \arg\max_i g_i(x, \emptyset) \qquad (13)$$

The first two approaches choose a model and sample it to completion, differing only in the order each model is considered. *Re-FORC (Highest Forecasted)* orders models by decreasing Gittins index; *Re-FORC (Smallest)* orders by model size, starting with the cheapest model $\pi_{small}$ and considering the most expensive last. After obtaining reward $R_{best}$ from one model, both approaches try the next model only if $g_i(x, \emptyset) > R_{best}$. The third approach, *Re-FORC Pandora*, applies Equation (12) directly, interleaving reasoning across models at the step level rather than committing to one model at a time.

## 4.4. Test-Time Scaling

Finally, we apply the Pandora's box greedy search framework to test-time scaling, where we allocate compute across multiple reasoning samples for a given model $\pi_i$. Each reasoning trace corresponds to a "box" in the Pandora's box problem (Section 4.2): opening a box means advancing a trace, and the agent must decide which trace to continue or whether to start a new one. We consider two strategies that differ in the granularity at which boxes are opened.

*Re-FORC-scaling.* Each complete trace is treated as a single open box. Given a query $x$ and completed reasoning traces (each sampled using Equation (11)), we draw a fresh sample

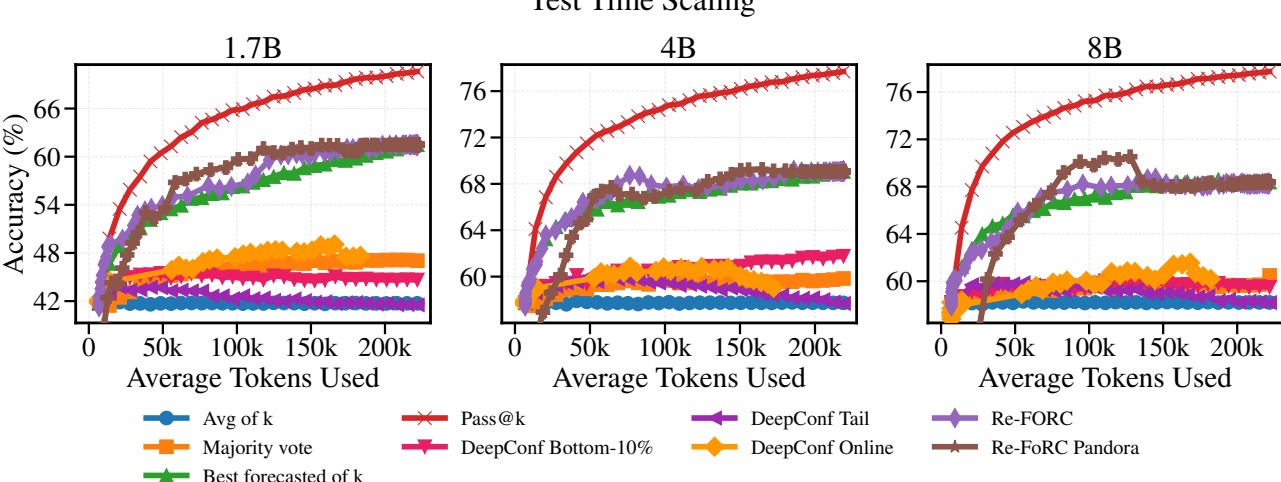

*Figure 4.* **Test-time scaling with Re-FORC.** Accuracy vs. average tokens used for Qwen3 1.7B (left), 4B (middle), and 8B (right), averaged across three competition-math benchmarks (AMC 2024, AIME 2024, AIME 2025); we sample $32\times$ per query. Per-dataset breakdowns in Appendix B.5. Re-FORC-scaling (Equation (14)) selects a model upfront via Equation (13) and resamples complete traces, while Re-FORC Pandora (Equation (15)) operates at the step level, dynamically selecting which model–trace pair to advance. Both methods are compared against repeated-sampling baselines (Avg-of-k, majority vote, best-forecasted-of-k), three DeepConf (Fu et al., 2026) variants (Online, Tail Conf, Bottom-10% Conf), and the oracle upper bound Pass@k. Best-forecasted-of-k uses the forecaster to select among $k$ completed samples. The DeepConf variants score traces using the mean of the lowest-decile (10%) of 512-token group confidences within a trace; *Online* gates generation in real time using this score, while *Bottom-10% Conf* filters completed traces and *Tail Conf* uses the mean confidence of the last 2,048 tokens instead. For the offline variants, we select using the highest score, and for the online we select using majority vote from the remaining traces. At peak compute, Re-FORC outperforms the strongest DeepConf variant (at any compute) by $+12.4$ pp ($+25.3\%$ relative) on 1.7B, $+7.3$ pp ($+11.8\%$) on 4B, and $+6.8$ pp ($+11.1\%$) on 8B. Re-FORC Pandora can outperform Re-FORC-scaling in the medium-to-high compute regime ($\geq 50$k tokens), while the simpler Re-FORC-scaling is preferable in the very low-compute regime ($\leq 25$k tokens).

if the Gittins index for a new trace exceeds the current best:

$$g_i(x, \emptyset) > R_{best} \tag{14}$$

This approach generates complete traces sequentially, deciding whether to resample after each step.

*Re-FORC Pandora scaling.* Rather than treating each trace as an atomic box, we apply the Pandora's box greedy search at the *step level*: each partial trace $z_j$ is a box that can be opened incrementally. Given a set of partial traces $\{z_1, \ldots, z_n\}$ for model $\pi_i$, at each step we select the trace with highest Gittins index and advance it by $\Delta$ tokens:

$$j^* = \arg\max_j \; g_i(x, z_j) \quad \text{s.t.} \quad g_i(x, z_{j^*}) > R_{best} \tag{15}$$

This is a direct application of Equation (12) where each model-trace pair is a box that we open one step at a time. The agent advances trace $z_{j^*}$ using model $\pi_i$ for $\Delta$ tokens, recomputes Gittins indices, and terminates when no continuation improves over the current best reward $R_{best}$. By interleaving exploration across partial trajectories at each step, Pandora scaling allocates compute where it is most promising rather than committing to full traces before deciding.

In Figure 4, we show results for all three Qwen-3 model sizes (1.7B, 4B, 8B) averaged across AIME 24/25 and

AMC 2024. Both Re-FORC-scaling (Equation (14)) and Re-FORC Pandora (Equation (15)) outperforms the majority-vote baseline across most compute budgets. In the high compute regime ($\geq$100k tokens, corresponding to many full-length samples per query), Re-FORC improves accuracy by 12.4 percentage points over DeepConf for the 1.7B model, 7.3 percentage points for the 4B, and 6.8 percentage points for the 8B. Notably, in the very low compute regime ($\leq$25k tokens), Pandora's step-level exploration incurs overhead from maintaining multiple partial traces, making it less efficient than the simpler Re-FORC-scaling—the benefits of multi-trajectory interleaving require sufficient compute budget to materialize.

## 5. Experiment details

### 5.1. Training Setup

We implement our adaptive reward forecaster as a lightweight adapter attached to pretrained reasoning models from the Qwen3 family (1.7B, 4B, 8B parameters) (Yang et al., 2025a). The base reasoning models remain frozen during forecaster training to preserve their reasoning capabilities while learning to predict future performance. The forecaster architecture consists of a self-attention pooling layer

that takes penultimate-layer activations $h_{1:n} \in \mathbb{R}^{n \times d}$ and aggregates sequence information into a fixed-size representation, followed by a linear projection head $g_\theta : \mathbb{R}^d \to \mathbb{R}^{2|\mathcal{T}|}$ that outputs Beta distribution parameters $(\alpha_t, \beta_t)$ for each time horizon $t \in \mathcal{T}$.

We use a uniform forecasting grid $\mathcal{T} = \{0, 512, 1024, \dots, 8192\}$ with linear interpolation for intermediate values, where Beta parameters are obtained via softplus activation to ensure positivity. The forecaster introduces minimal computational overhead, requiring only a single forward pass through the base model during training to extract activations, amortizing forecasts over all horizons without additional thinking tokens during inference.

Training data is generated by sampling problem instances $(x_i, y_i)$ from DeepScaleR-Preview (Luo et al., 2025) and creating full unconstrained reasoning trajectories up to a maximum thinking length of 8192 tokens. We extract partial traces at regular intervals and use Monte Carlo estimation with $N = 8$ samples to compute empirical success rates, with rewards clipped to $(\varepsilon, 1 - \varepsilon)$ where $\varepsilon = 10^{-6}$ for numerical stability.

### 5.2. Evaluation Setup

We evaluate on four mathematics reasoning datasets: AMC 2024 (Art of Problem Solving, 2024f;e;d;c), Math500 (Lightman et al., 2024), and AIME 2024/25 (Art of Problem Solving, 2024a;b; 2025a;b). Forecasting performance is measured using Pearson correlation ($\rho$), mean squared error (MSE), and mean absolute error (MAE) between predicted and true reward values, while compute-aware inference is evaluated on accuracy-compute trade-offs measuring both final accuracy and total thinking tokens consumed. We use a maximum number of thinking tokens of 8192. We use 32 samples per problem for evaluation, unless otherwise stated.

Our experimental comparisons include unconstrained generation as a standard reasoning baseline without early stopping, fixed token limits representing simple cutoffs without adaptive decision-making, single-model baselines using only the largest or smallest available model, and oracle routing with ground-truth access (Pass@k) as a theoretical upper bound. These baselines allow us to isolate the contributions of adaptive forecasting versus simpler heuristic approaches.

*Baselines.* For each application, we compare Re-FORC against the baselines most appropriate to that setting. **Early stopping** is compared against S1 (Muennighoff et al., 2025), which uses fixed-length token cutoffs, and DeepConf (Fu et al., 2026), a recent confidence-based adaptive reasoning method, alongside unconstrained generation as a no-stopping reference. **Test-time scaling** is additionally compared against majority voting (Wang et al., 2023b), average-

of-$k$ sampling, best-forecasted-of-$k$ (which selects the trace with the highest forecasted reward without adaptive stopping), three DeepConf variants (Online, Tail Conf, Bottom-10% Conf), and Pass@$k$ as the oracle upper bound with ground-truth access. **Model selection** is compared against running each individual model in isolation (All 1.7B, All 4B, All 8B), Avg-of-$N$, and Pass-of-$N$. Together, these baselines isolate the contribution of explicit reward forecasting from simpler heuristic and confidence-based approaches.

## 6. Results

*User-controlled inference:* Users can dynamically control computational expenditure by selecting an appropriate $\lambda$ value at inference time based on their accuracy and cost requirements. Re-FORC then automatically optimizes the reward-compute trade-off, as demonstrated in Figure 2, Figure 3, and Figure 4. The method proves especially advantageous in intermediate compute regimes, where users desire meaningful quality improvements without prohibitive costs—consistently achieving superior accuracy compared to baselines at equivalent compute budgets. For instance, in early stopping (Figure 2; per-dataset breakdown in Appendix B.2), the maximum accuracy advantage of Re-FORC over S1 at equal compute is $+3.8$ pp at 1.5k tokens for the 1.7B model, $+4.2$ pp at 2k tokens for the 4B, and $+3.9$ pp at 4k tokens for the 8B—larger models benefit from Re-FORC most at higher token budgets. Similarly, in test-time scaling (Figure 4), Re-FORC with the 8B model provides maximum improvements in the 100k-token range ($100k \approx 32 \times 3k$, since we sample $32 \times$ per query).

*Compute-aware applications:* Our experiments demonstrate the effectiveness of Re-FORC across three key applications. First, Re-FORC-stopping (Eq. 11) provides smooth accuracy-compute frontiers, where moderate $\lambda$ values preserve most peak accuracy while reducing reasoning tokens. Re-FORC-stopping reduces compute by 24% on average compared to S1 and 16% compared to DeepConf for the Qwen3 4B model, while maintaining accuracy. Second, Re-FORC-Pandora (Eq. 13) outperforms the largest model alone, reaching 1.7 percentage points higher peak accuracy while needing 12% less compute to match its accuracy. Finally, Re-FORC Pandora applied to Test Time Scaling (Eq. 14) achieves superior accuracy-compute trade-offs compared to best forecasted in the medium or high compute regime.

*Flexible base model training:* Re-FORC maintains complete independence from the base reasoning model's training procedure, enabling practitioners to optimize the underlying model using any algorithm suited to their application. This stands in contrast to methods such as L1 (Aggarwal & Welleck, 2025) or e1 (Kleinman et al., 2025) that require modifying model weights through training. By oper-

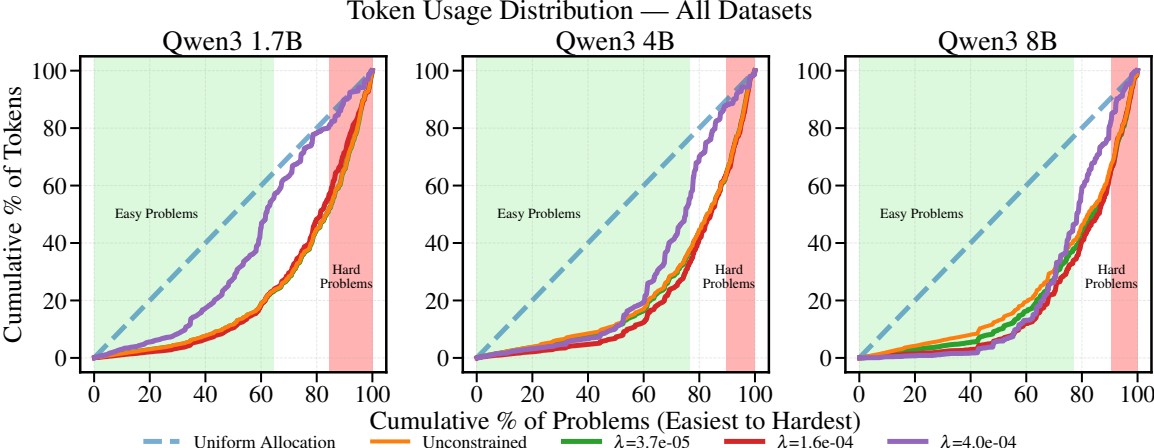

*Figure 5.* **Token distribution and problem difficulty for Qwen3 models combined across datasets.** Problems are combined across datasets (MATH500, AMC 2024, AIME 2024, AIME 2025) and ordered by per-model solve rate; "easy" problems are solved in $\geq 90\%$ of trials and "hard" ones in $<50\%$. Each curve shows the cumulative percentage of total tokens spent versus the cumulative percentage of problems, with the dashed diagonal indicating uniform allocation. Increasing $\lambda$ represents a higher cost sensitivity, encouraging more selective compute use. At high $\lambda = 4.0 \times 10^{-4}$, the models allocate a smaller proportion of compute to the more difficult problems compared to smaller values of $\lambda$ and unconstrained generation. For the 8B model, on the easiest $\sim 40\%$ of problems, increasing $\lambda$ leads to a smaller percentage of compute used on such problems. Per-dataset results are in Appendix B.4.

ating solely at inference time, our approach traces the entire accuracy-compute trade-off curve while providing fine-grained cost control—all without altering the base model's training or architecture.

*Difficulty-based allocation.* We further analyze how Re-FORC-stopping (Eq. 11 and Figure 2) – which depends on future reward forecasts, current best reward, and cost sensitivity $\lambda$ – leads to a reduction in tokens. We order problems by solve rate and plot the cumulative percentage of tokens used versus the cumulative percentage of problems (Figure 5). At high $\lambda = 4.0 \times 10^{-4}$, the token allocation curves differ compared to unconstrained generation. Across model sizes, there is a smaller proportion of tokens allocated to the most difficult problems. For the 8B model, the proportion of compute is also reduced on the easiest problems, whereas for the 1.7B model, the proportion increases.

*Forecast accuracy improves with reasoning progress:* Forecast quality improves as chain-of-thought tokens increase, with higher $\rho$ and lower MSE/MAE (see Figure 1).

## 7. Conclusion

We introduced Re-FORC, an adaptive reward prediction approach that enables efficient control of compute (both model size and reasoning length) over chain-of-thought reasoning by thresholding the forecasting functional using the Gittins index policy. We formulate the reward-compute prediction problem using Pandora's box greedy search (Weitzman, 1979; Scully & Terenin, 2025) and provide empirical techniques to approximate the Gittins index policy for reasoning models (Achille & Soatto, 2026) in practice. Our method trains lightweight forecasters (adapters) on top of frozen rea-

soning models to predict future reward-token trade-off (reasoning trajectory outcomes). Our forecaster enables three key inference-time applications: (1) early stopping of unpromising reasoning trajectories, (2) compute-aware model selection from a pool of reasoning models, and (3) cost-aware test-time scaling. Results across four math benchmarks demonstrate that forecaster-guided strategies consistently outperform baseline approaches, achieving superior accuracy-compute trade-offs.

**Limitations.** Collecting forecaster training data is computationally expensive, requiring multiple reasoning trajectories per query at various length intervals, with costs scaling with both dataset size and model capacity. Additionally, the forecaster occasionally exhibits overconfidence, predicting higher rewards than the base model can realistically achieve, which can lead to wasteful computation by continuing to sample rather than terminating early. Extended training with larger datasets can help mitigate this, though complete calibration remains an ongoing challenge.

## Impact Statement

This work aims to improve the computational efficiency of reasoning in large language models, potentially reducing energy consumption and making advanced AI capabilities more accessible. By enabling adaptive compute allocation, Re-FORC allows systems to avoid wasteful computation on queries that can be solved with less effort, while appropriately investing resources in more challenging problems. This could democratize access to high-quality AI reasoning by reducing costs.

However, more efficient reasoning systems could also accelerate automation in knowledge work, with potential labor market implications. Additionally, while our method improves efficiency, it does not address the underlying correctness or safety of the reasoning models themselves—a miscalibrated forecaster could prematurely terminate reasoning on important queries or over-allocate compute to adversarial inputs.

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

# Appendix

# A. Theoretical Derivations

In this section, we provide derivations for the key equations used in our compute-aware inference framework.

## A.1. Gittins Index Closed Form

We derive the closed-form Gittins index (Equation (3)) from its definition (Equation (2)) for binary rewards.

**Proposition A.1.** *For binary rewards $R \in \{0, 1\}$ with success probability $p = P(R = 1)$, the Gittins index satisfying $\mathbb{E}[(R - g)_+] = \lambda t$ has closed form $g = 1 - \frac{\lambda t}{p}$ when $p > \lambda t$.*

*Proof.* For binary $R \in \{0, 1\}$:

$$\begin{aligned}
\mathbb{E}[(R - g)_+] &= P(R = 1) \cdot (1 - g)_+ + P(R = 0) \cdot (0 - g)_+ \\
&= p \cdot \max(1 - g, 0) + (1 - p) \cdot \max(-g, 0) \\
&= p(1 - g) \quad \text{for } g \in [0, 1]
\end{aligned} \tag{16}$$

Setting this equal to $\lambda t$ per the Gittins definition:

$$\begin{aligned}
p(1 - g) &= \lambda t \\
1 - g &= \frac{\lambda t}{p} \\
g &= 1 - \frac{\lambda t}{p}
\end{aligned} \tag{17}$$

This is valid when $p > \lambda t$, ensuring $g > 0$ (positive reservation value). $\qquad\square$

## A.2. Optimal Horizon Maximizes Gittins Index

We show that the optimal compute budget (Equation (4)) maximizes the Gittins index over possible horizons.

**Proposition A.2.** *The optimal horizon $t^* = \arg\min_t \frac{\lambda t}{\psi(t)}$ equivalently maximizes the Gittins index $g(t) = 1 - \frac{\lambda t}{\psi(t)}$.*

*Proof.* Since $g(t) = 1 - \frac{\lambda t}{\psi(t)}$, we have:

$$\begin{aligned}
\arg\max_t g(t) &= \arg\max_t \left( 1 - \frac{\lambda t}{\psi(t)} \right) \\
&= \arg\min_t \frac{\lambda t}{\psi(t)} = t^*
\end{aligned} \tag{18}$$

Thus selecting $t^*$ that minimizes the cost-to-success ratio equivalently maximizes the reservation value. $\qquad\square$

## A.3. Expected Improvement for Binary Rewards

We show how the expected improvement (Equation (9)) simplifies for binary rewards.

**Proposition A.3.** *For binary rewards $R \in \{0, 1\}$ with $P(R = 1) = \psi(t \mid x, z, \pi)$ and current best reward $R_{best}$, the expected improvement is:*

$$\Delta_t J = \psi(t \mid x, z, \pi)(1 - R_{best}) - \lambda t \tag{19}$$

*Proof.* Starting from the general expected improvement:

$$\begin{aligned}
\mathbb{E}[(R - R_{best})_+] &= P(R = 1) \cdot (1 - R_{best})_+ + P(R = 0) \cdot (0 - R_{best})_+ \\
&= \psi(t \mid x, z, \pi) \cdot (1 - R_{best}) + 0 \\
&= \psi(t \mid x, z, \pi)(1 - R_{best})
\end{aligned} \tag{20}$$

where we use $(1 - R_{best})_+ = 1 - R_{best}$ since $R_{best} \in [0, 1]$, and $(0 - R_{best})_+ = 0$ since $R_{best} \geq 0$. Subtracting the cost term yields the result. $\qquad\square$

### A.4. Equivalence of Stopping Conditions

We show that the expected improvement condition $\Delta_t J > 0$ is equivalent to the Gittins threshold rule $g > R_{best}$.

**Proposition A.4.** *For binary rewards, continuing reasoning is beneficial ($\Delta_t J > 0$) if and only if the Gittins index exceeds the current best reward ($g > R_{best}$).*

*Proof.* Starting from the condition for positive expected improvement:

$$\begin{aligned}
\Delta_t J > 0 &\iff \psi(t)(1 - R_{best}) - \lambda t > 0 \\
&\iff \psi(t)(1 - R_{best}) > \lambda t \\
&\iff 1 - R_{best} > \frac{\lambda t}{\psi(t)} \\
&\iff 1 - \frac{\lambda t}{\psi(t)} > R_{best} \\
&\iff g > R_{best}
\end{aligned} \tag{21}$$

where we use the closed-form Gittins index $g = 1 - \frac{\lambda t}{\psi(t)}$ from the previous derivation. $\square$

## B. Additional Experiments

### B.1. Forecasting performance per dataset

We show the per-dataset forecasting performance corresponding to Figure 1 in the main paper. Each panel reports Pearson $\rho$ correlation $\rho$, MSE, and MAE between predicted and true reward as the chain-of-thought progresses, separately for AIME 2024, AIME 2025, AMC 2024, and MATH500.

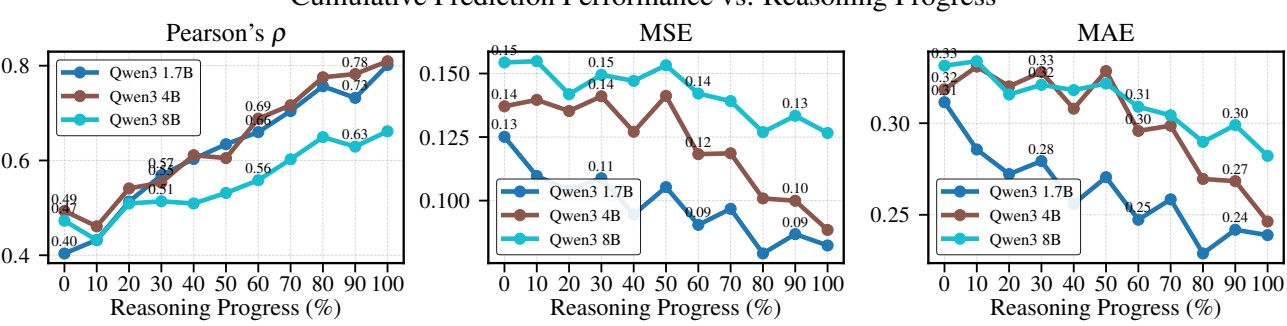

*Figure 6.* Forecasting performance on AIME 2024.

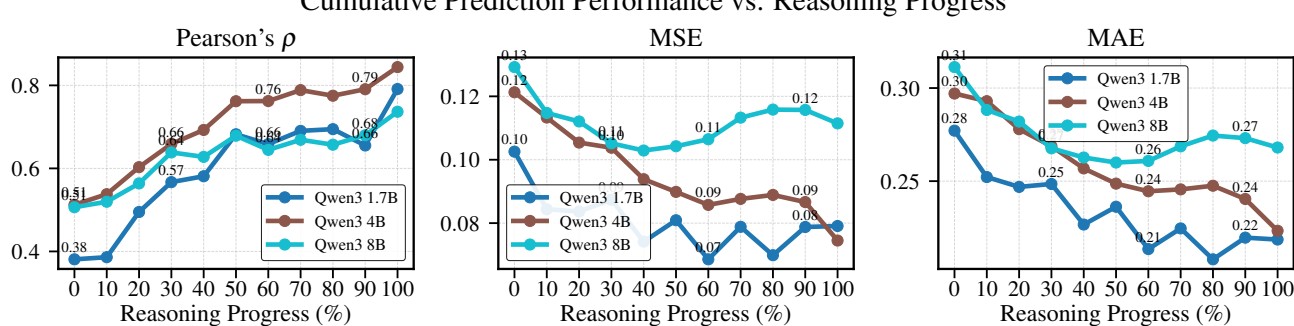

*Figure 7.* Forecasting performance on AIME 2025.

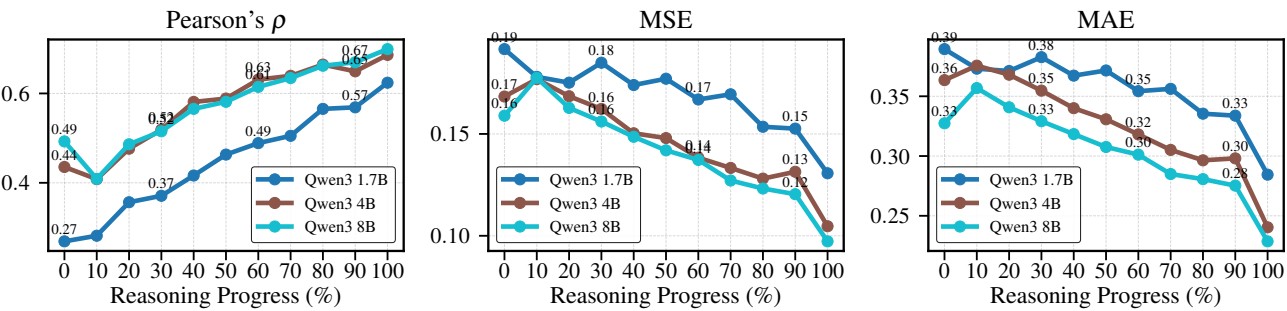

*Figure 8.* Forecasting performance on AMC 2024.

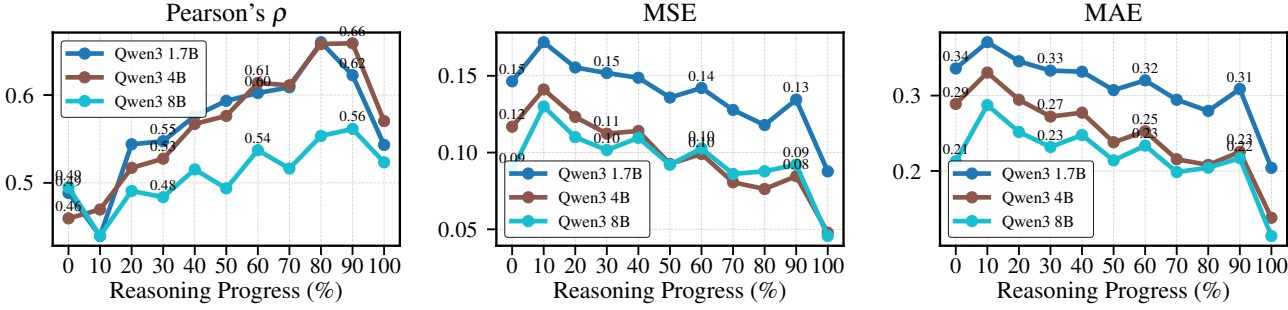

*Figure 9.* Forecasting performance on MATH500.

## B.2. Early stopping per dataset

Per-dataset breakdown of the early-stopping experiment in Figure 2. Each figure shows accuracy vs. tokens for Qwen3 1.7B, 4B, and 8B side by side, comparing Re-FORC against S1, DeepConf, and unconstrained generation.

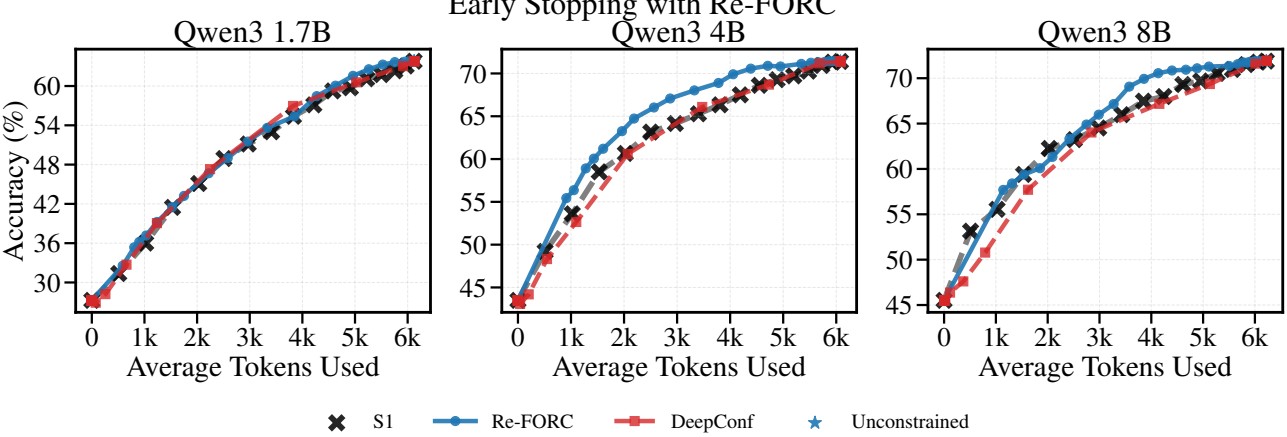

*Figure 10.* Early stopping on AMC 2024.

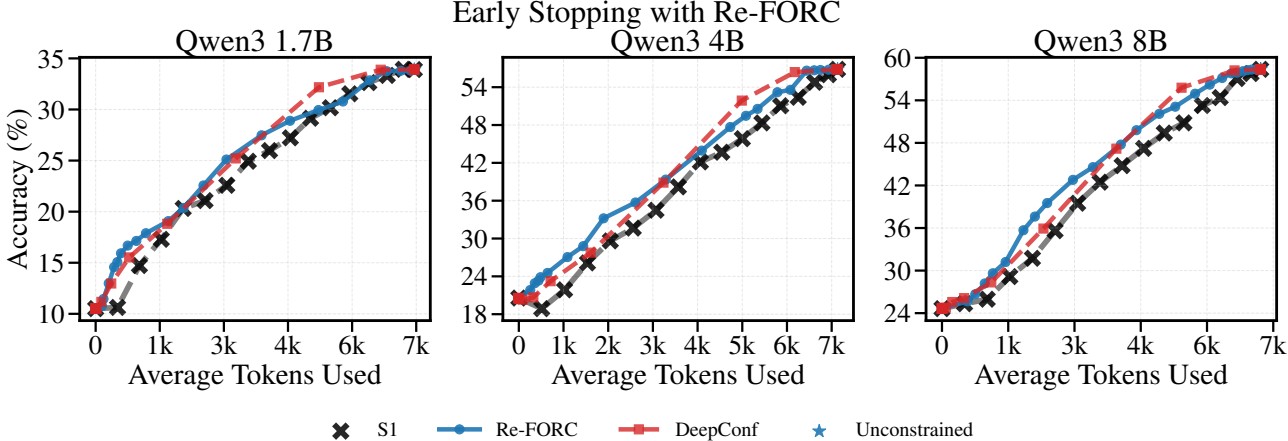

*Figure 11.* Early stopping on AIME 2024.

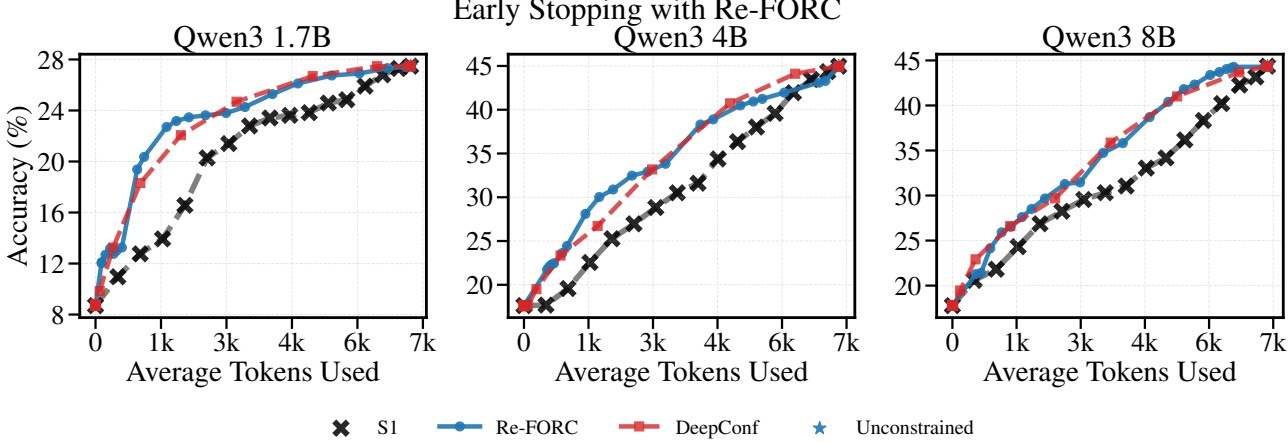

*Figure 12.* Early stopping on AIME 2025.

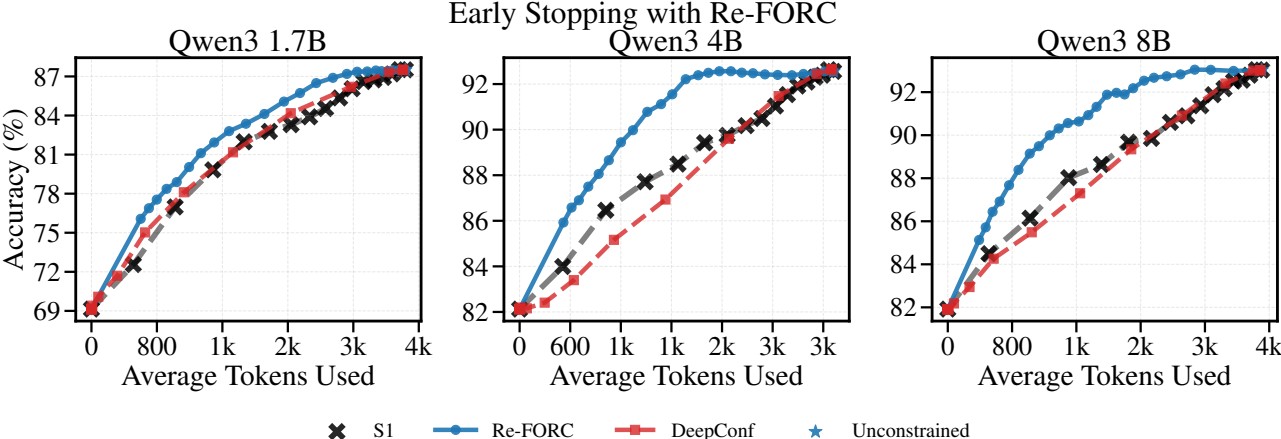

*Figure 13.* Early stopping on MATH500.

## B.3. Model selection per dataset

Per-dataset breakdown of the model-selection experiment from Figure 3. Each figure shows accuracy vs. average compute (timesteps × T-FLOPs) on a single benchmark, comparing the three Re-FORC variants (Smallest-First, Highest-Forecasted-First, Pandora) against the individual-model anchors (All 1.7B / 4B / 8B), Pass-of-N (oracle), and Avg-of-N.

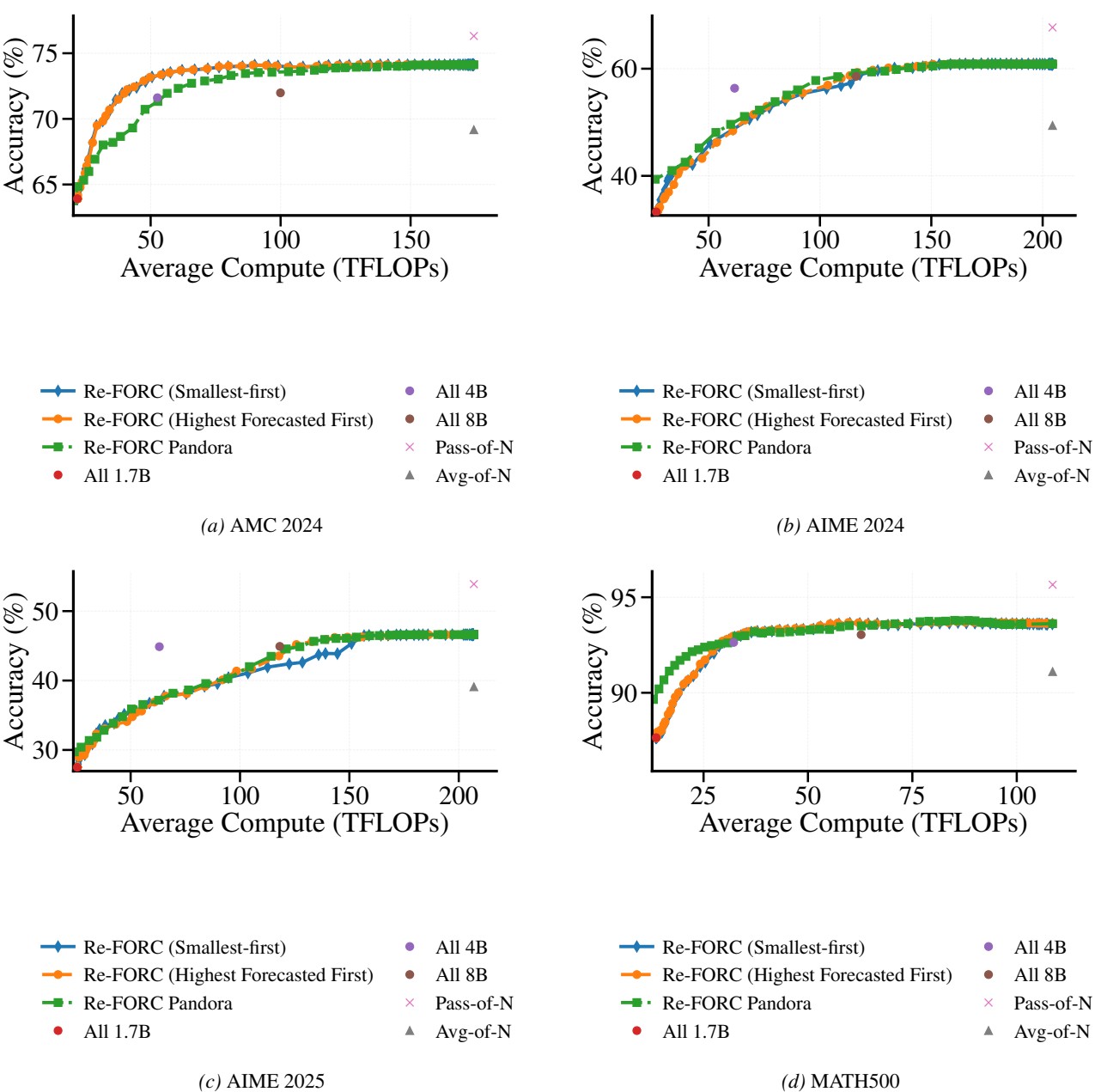

*Figure 14.* Per-dataset cost-aware model selection across the four math benchmarks.

## B.4. Token-usage distribution per model size

Per-model-size, per-dataset breakdown of the token-distribution figure in Figure 5. Each subfigure shows the cumulative share of tokens spent versus cumulative problem difficulty for one Qwen3 model size, broken out across the four math benchmarks (AMC 2024, AIME 2024, MATH500, AIME 2025). Problems are ordered by difficulty from easiest to hardest based on solve rate. The dashed diagonal indicates uniform allocation; increasingly convex curves indicate selective compute use that concentrates effort on harder problems. On difficult datasets (like AIME 2024 or AIME 2025) the models preferentially allocates computation to easier problems for high $\lambda=4.0\times10^{-4}$. In contrast for an easier dataset (Math500), the curves for 8B model are increasingly convex for increasing $\lambda$ (allocating a smaller fraction of compute to easier problems than to harder ones).

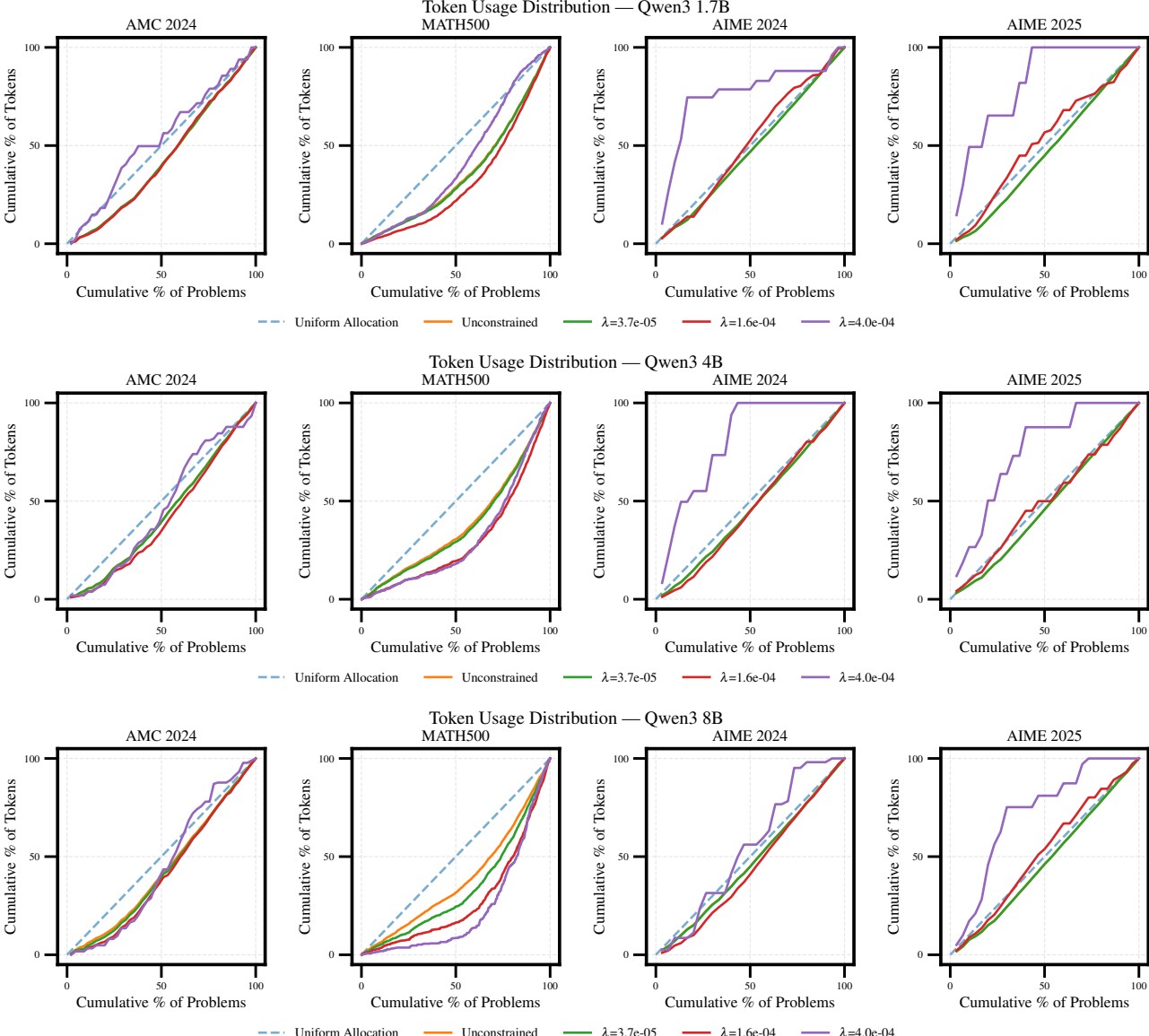

*Figure 15.* Token-usage distribution per dataset for the 1.7B Model (Top Row), 4B model (Middle Row) and 8B model (Bottom row).

## B.5. Test-time scaling per dataset

Per-dataset breakdown of the test-time-scaling experiment in Figure 4. Each figure shows accuracy vs. average tokens used for Qwen3 1.7B (left), 4B (middle), and 8B (right) on a single benchmark. Baselines include Avg-of-$k$, majority vote, best-forecasted-of-$k$, three DeepConf variants (Online, Tail, Bottom-10%), and Pass@$k$ as an oracle upper bound.

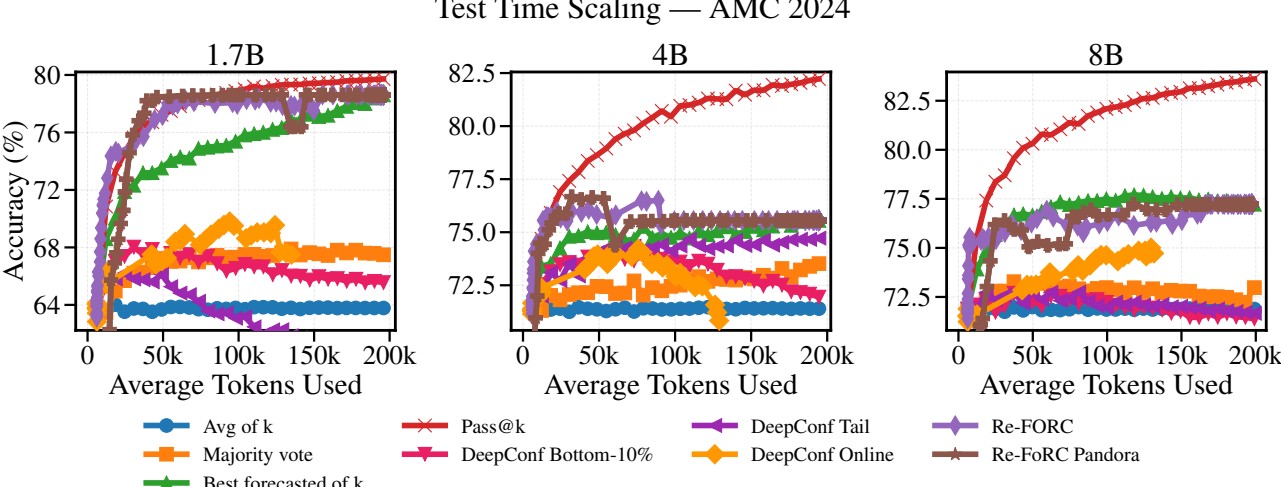

*Figure 16.* Test-time scaling on AMC 2024.

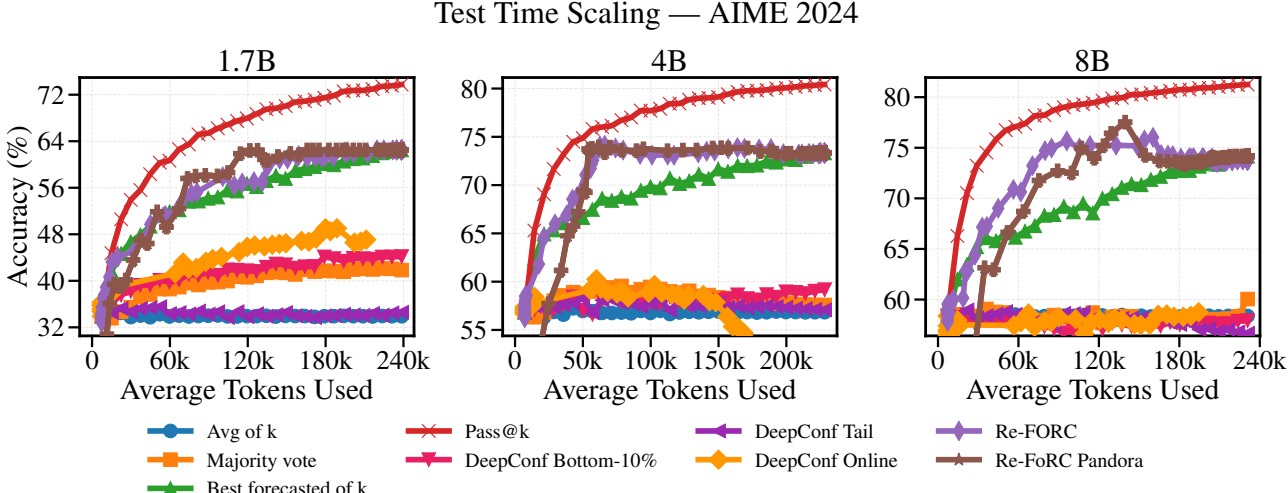

*Figure 17.* Test-time scaling on AIME 2024.

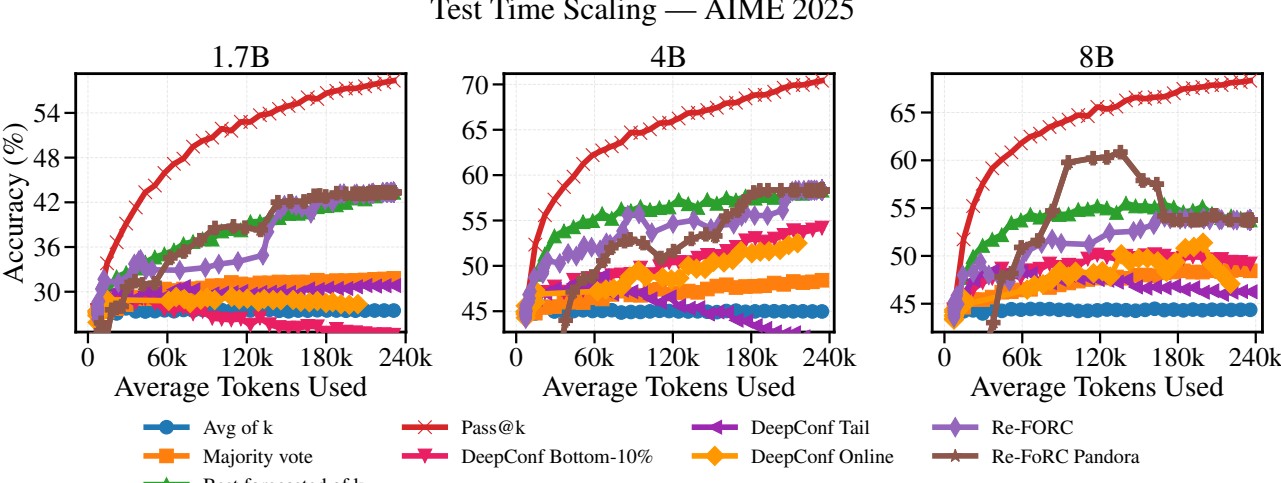

*Figure 18.* Test-time scaling on AIME 2025.

### B.6. Out-of-distribution test-time scaling on MMLU-Pro

To probe whether a forecaster trained on one domain can be applied on an unseen domain, we evaluate Re-FORC on MMLU-Pro (Wang et al., 2024) using a forecaster trained only on DeepScaleR-Preview math data, with no MMLU-Pro fine-tuning. We evaluate on 100 problems, sampling $k = 8$ traces per problem for all three Qwen3 model sizes (1.7B, 4B, 8B). Figure 19 shows the full test-time-scaling curves and Table 1 reports the peak accuracy reached by each baseline method along its curve and Re-FORC accuracy at maximum compute.

Re-FORC ties or beats every baseline on 4B and 8B: it ties DeepConf Tail at $39.5\%$ on 4B (best DeepConf variant) and leads the best DeepConf variant by $+0.8$ pp on 8B ($40.3\%$ vs. $39.5\%$). On 1.7B the picture is reversed—DeepConf Tail reaches $36.3\%$, while Re-FORC, Re-FORC Pandora, and Majority Vote cluster near $33.8\%$.

We caveat the results: MMLU-Pro contains reasoning questions across different domains (including mathematics), and so may share similarities with the DeepScaleR training data. Training the forecaster on a broader task mix is likely important for better applicability across domains.

| Model | Re-FORC | Re-FORC Pandora | DeepConf Online | DeepConf Tail | DeepConf Bottom-10% | Majority | Pass@$k$ |
|-------|---------|-----------------|-----------------|---------------|---------------------|----------|----------|
| 1.7B  | 33.8%   | 33.8%           | 34.3%           | **36.3%**     | 34.3%               | 33.8%    | 41.3%    |
| 4B    | **39.5%** | 39.5%         | 38.7%           | **39.5%**     | 38.7%               | 39.2%    | 41.3%    |
| 8B    | **40.3%** | 40.3%         | 38.8%           | 39.5%         | 39.3%               | 38.8%    | 43.0%    |

*Table 1.* **Out-of-distribution test-time scaling on MMLU-Pro** with a math-only forecaster: peak accuracy reached by each baseline method along its test-time-scaling curve compared against Re-FORC accuracy at maximum compute. Re-FORC ties or beats every non-oracle baseline for the 4B and 8B models; DeepConf Tail is strongest for 1.7B. Pass@$k$ is the oracle upper bound.

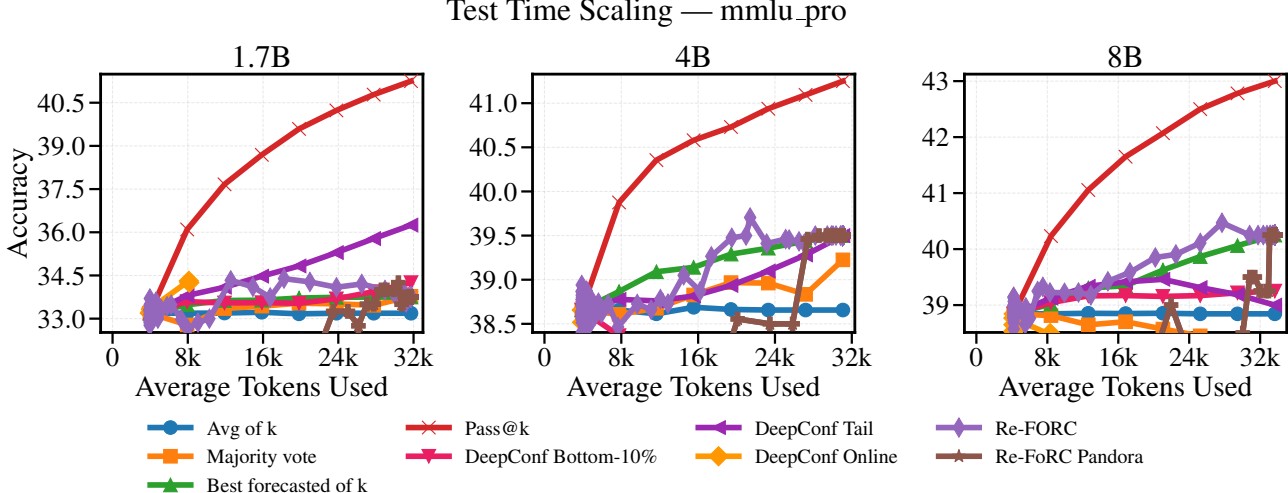

*Figure 19.* **Out-of-distribution test-time scaling on MMLU-Pro.** Accuracy vs. average tokens used for Qwen3 1.7B (left), 4B (middle), and 8B (right) using a math-only forecaster (no MMLU-Pro fine-tuning). All baselines (Avg-of-$k$, majority vote, best-forecasted-of-$k$, three DeepConf variants, Pass@$k$) are shown alongside Re-FORC and Re-FORC Pandora.

