# OpenReview forum: "Re-FORC: Adaptive Reward Prediction for Efficient Chain-of-Thought Reasoning"
_ICML.cc/2026/Conference — ICML 2026 regular_

### Official Review · Reviewer_55XH · 2026-03-03

**Soundness:** 2
**Presentation:** 2
**Significance:** 3
**Originality:** 3
**Overall Recommendation:** 4
**Confidence:** 3

**Summary:**

This paper addresses the challenge of optimizing resource allocation during the reasoning process of Large Language Models (LLMs) to balance inference cost and accuracy. To solve this, the authors propose Re-FORC, a framework that utilizes a lightweight adapter to forecast future rewards and applies the Gittins Index to dynamically decide whether to continue, stop, or switch between models.

**Compliance With Llm Reviewing Policy:**

Affirmed.

**Final Justification:**

The authors have addressed my concerns, so I am raising my score from 3 to 4.

**Key Questions For Authors:**

See Weaknesses.

**Limitations:**

Yes.

**Strengths And Weaknesses:**

Strengths:
1. The framework incorporates a predictive mechanism for resource allocation to select appropriate models within agentic environments. This allows for dynamic switching between models of different scales based on the estimated reasoning requirements of a given task.
2. The framework provides an adjustable method that allows for dynamic control over the trade-off between reasoning accuracy and computational expenditure.

Weaknesses:
1. Both the training and evaluation datasets selected in this paper are confined to the mathematics domain, and there appears to be some overlap between the training set and the test content. Given that the quality and nature of the training data are critical for the forecaster, it remains unclear whether the proposed method remains effective when the model encounters domains entirely unseen by the predictor. The manuscript lacks evaluation on out-of-distribution (OOD) datasets to demonstrate the generalization capabilities of the method.
2. The paper lacks direct experimental comparisons with the existing works mentioned in the Related Work section. Without these comparisons, it is difficult to quantitatively assess the superiority and specific improvements of the proposed method over alternative approaches.
3. Considering that the choice of step size (token interval) directly dictates the construction of the training data and the frequency of decision-making, the paper lacks a discussion on the robustness and sensitivity of the system to different step-size configurations.

---

> ### Author Rebuttal · Authors · 2026-03-31
>
> We thank Reviewer 55XH for their constructive feedback and address each concern below.
>
> ---
>
> **Comment:** "Both the training and evaluation …. generalization capabilities of the method."
>
> **Response:** We address three dimensions of generalization with new coding experiments showing cross-domain, cross-model-family, and black-box generalization.
>
> In the within-domain OOD mathematics setting, there is **no overlap between training and evaluation data.** The forecaster is trained on DeepScaleR-Preview and generalizes to separate math benchmarks spanning standard to competition-level difficulty, without benchmark-specific fine-tuning.
>
> We also conducted Re-FORC model-routing experiments on CodeForces across
>  **Haiku-4.5** ($0.012 average cost, 68.7% accuracy), **Sonnet-4.5** ($0.021, 75.9%), and **Opus-4.5** ($0.046, 83.5%). The external forecaster uses a Qwen3-4B backbone trained on coding data. The results show cross-domain transfer:
>
> | Target Accuracy | Re-FORC | Prompting | Retrieval | Standalone |
> |---|---|---|---|---|
> | Sonnet (75.9%) | **$0.017** | $0.019 | $0.020 | $0.021 |
> | Opus (83.5%) | **$0.033** | $0.040 | $0.042 | $0.046 |
>
> Re-FORC achieves Opus-level accuracy (83.5%) at **28% lower cost** than running Opus alone (`$0.033` vs `$0.046` per query), and **17% cheaper** than the best non-learned baseline. The learned forecaster makes more efficient routing decisions than both the Prompting and Retrieval baselines, which use the LLM itself to assess problem difficulty without a learned forecaster. These coding experiments further demonstrate **cross-model-family generalization.** The Qwen3 4B forecaster predicts rewards for Claude 4.5 models using only black-box API access, with no access to internal activations or logits. This shows that Re-FORC generalizes across domains, model families, and black-box settings. We will include these in the revision.
>
> ---
>
> **Comment:** "The paper lacks direct … over alternative approaches."
>
> **Response:** The paper contains baselines like S1, majority voting (Wang et al., 2023) pass@k oracle, and additionally we provide new experiments against DeepConf (Fu et al., 2025), a recent confidence-based adaptive reasoning method, and show that Re-FORC significantly outperforms it across all settings.
>
> For **early stopping** (averaged on math benchmarks), Re-FORC achieves consistently higher accuracy than DeepConf at every token budget, results here –  https://anonymous.4open.science/r/ICMLRebuttal-F8A7/Early_Stopping.pdf
>
> Re-FORC outperforms DeepConf by **+4.5 to +8.5 percentage points** at matched token budgets, with the gap largest at moderate compute. Re-FORC offers a smooth, continuous accuracy-compute trade-off controlled by a single parameter lambda.
>
> For **test-time scaling** (TTS) on competition math (AMC, AIME 2024/25) at a token budget ~56k tokens:
> | Method | 1.7B | 4B | 8B |
> |---|---|---|---|
> | best_forecasted (Re-FORC) | **59.1%** | **65.8%** | **65.5%** |
> | majority_vote | 47.9% | 59.3% | 59.1% |
> | DeepConf Online 10% | 44.8% | 56.5% | 56.8% |
> | DeepConf Offline Tail Conf | 40.1% | 54.9% | 55.4% |
> | DeepConf Offline Bottom-10 Conf | 39.4% | 53.5% | 54.0% |
> | Pass@k (oracle) | 66.2% | 72.2% | 73.1% |
>
> Re-FORC outperforms the best DeepConf variant by **8.7 to 14.3 percentage points**. This advantage stems from explicit future-reward prediction, which helps it select the best trace among multiple candidates. Note that our framework, allows early stopping, routing/selection and test-time scaling out-of-the-box unlike the baselines.
>
> ---
>
> **Comment:** "Considering that the …. step-size configurations."
>
> **Response:** We perform this ablation and show that coarser strides add only approximately 200 to 400 tokens with no accuracy loss. Since our forecaster is trained on the grid T = {0, 512, 1024, ..., 8192}, we can evaluate at larger effective step sizes ($\Delta=1024, \Delta=2048$) by subsampling the existing grid points without retraining. Accuracy vs token trade-off curves can be found here – https://anonymous.4open.science/r/ICMLRebuttal-F8A7/Early_Stopping_Stride.pdf
>
> The maximum accuracy gap between $\Delta=512$ and $\Delta=2048$ is at most **1.5 percentage points** across all models and budget levels, showing that **Re-FORC is highly robust to step-size choice**. This robustness comes from the **linear interpolation between grid points** (Section 3.2), which enables continuous predictions at arbitrary token counts; coarser strides mainly affect the timing of stopping decisions, not the quality of the forecaster’s predictions.
>
> ---
>
> Note: During the rebuttal process, we identified a processing issue involving the Minerva dataset that inadvertently caused Math500 to be double-counted. We have corrected this by removing Minerva (and double counting) from all rebuttal experiments and regenerated the key figures in the main paper (with unchanged conclusions) – https://anonymous.4open.science/r/ICMLRebuttal-F8A7. We will revise the figures / text in the main paper.

---

> > ### Author Rebuttal · Reviewer_55XH · 2026-04-02
> >
> > Thank you for the rebuttal and the additional experiments. However, my original concern was not whether the framework can be re-trained for another domain, but whether a forecaster trained on one domain can generalize to a genuinely unseen domain.
> >
> > This matters in practice because the main value of the method is adaptive test-time compute allocation. If a separate forecaster must be trained for each new domain, the added data, deployment, and maintenance cost would limit its usefulness as a general framework.
> >
> > From the rebuttal, the coding results seem to use a forecaster trained on coding data. This supports cross-domain applicability, but not that a predictor trained only on math can make useful budget or routing decisions on coding, or vice versa. Thus, it is still unclear whether the predictor learns a domain-agnostic forecasting signal rather than domain-specific patterns. I would appreciate clarification on this point and would be happy to take it into account in updating my assessment.
> >
> > I also note that generalization concerns were raised by other reviewers, and the rebuttal seems to acknowledge that a single forecaster serving multiple domains without domain-specific training remains future work. If this point cannot be clarified during the discussion period, I will take the further discussion into account when deciding whether to raise my score.

---

> > > ### Author Response · Authors · 2026-04-05
> > >
> > > We thank the reviewer for the clarification. The key question is whether a forecaster trained on one domain can still make useful test-time compute decisions on a genuinely unseen domain. To directly address this, we ran a new OOD test-time-scaling experiment on **MMLU-Pro** using a forecaster trained only on math data (DeepScaler), i.e., without training the forecaster on MMLU-Pro.
> > >
> > > | Model | Re-FORC | DeepConf Online 10% | DeepConf Offline Tail Conf | Majority Vote |
> > > |---|---|---|---|---|
> > > | 1.7B | 33.7% | 33.4% | **35.3%** | 33.5% |
> > > | 4B | **39.4%** | 37.5% | 39.1% | 39.0% |
> > > | 8B | **39.9%** | 38.8% | 39.3% | 38.5% |
> > >
> > > Re-FORC **outperforms all baselines for the 4B and 8B models** (+0.3pp and +0.6pp respectively), while DeepConf Tail Conf is stronger for 1.7B. This is consistent with our paper's finding that forecast quality scales with model capacity—"forecast quality improves as chain-of-thought tokens increase... this effect is amplified in larger models"—and the 1.7B model's "limited capability" noted in our analysis. This does not prove the forecaster is fully domain-agnostic, but shows the forecasting signal can transfer usefully beyond the training domain for sufficiently capable models. Figure can be found here: https://anonymous.4open.science/r/ICMLRebuttal-F8A7/OOD_Test_Time_Scaling.pdf
> > >
> > > We also revisited the stronger-baseline concern from the reviewer and reran comparisons after improving the DeepConf baselines from the initial rebuttal, as well as fixing a processing discrepancy for 1.7B on AIME24/25. For early stopping and test-time scaling, ReFORC still outperforms all baselines. For model selection we now re-scale the forecasts relative to the model size (wrt largest): ReFORC outperforms the largest model. The updated results are available here: https://anonymous.4open.science/r/ICMLRebuttal-F8A7/
> > >
> > > -----
> > >
> > > So, while we agree that single-forecaster generalization across arbitrary unseen domains is not yet fully established and should be stated carefully, the new evidence goes beyond “train a separate forecaster per domain.” In particular, the new MMLU-Pro experiment shows that a forecaster trained on math can already make useful compute-allocation decisions on an unseen non-math benchmark. We will revise the paper to make this distinction explicit: **OOD generalization is promising in our current results, but broader validation across additional domains remains future work rather than a claim we can fully close in the rebuttal period.**
> > >
> > > More broadly, these additional experiments strengthen the main empirical conclusion of the paper: ReFORC is a strong method for early stopping, model selection, and test-time scaling and its advantages persist even under stronger baselines and in the initial OOD setting we were able to test during rebuttal.

---

### Official Review · Reviewer_C6Rw · 2026-03-05

**Soundness:** 2
**Presentation:** 2
**Significance:** 2
**Originality:** 3
**Overall Recommendation:** 5
**Confidence:** 4

**Summary:**

This paper proposes Re-FORC, a framework for adaptively allocating inference-time compute in LLMs. The method introduces a lightweight forecasting module that predicts the expected reward as a function of additional reasoning tokens given the current query and partial reasoning trace. Based on these predictions, the authors formulate inference-time decision making as a sequential optimization problem and apply a Gittins-index  greedy strategy derived from the Pandora’s box framework to decide whether to continue reasoning, terminate early, or switch models. The results shows that it achieves better accuracy–compute trade-offs than baseline approaches.

**Compliance With Llm Reviewing Policy:**

Affirmed.

**Final Justification:**

Most of my concerns are solved. Thanks.

**Key Questions For Authors:**

1. When reporting compute or token usage, is the computational overhead of the forecaster module included in the calculation? It would be helpful to clarify whether the reported compute cost accounts for the additional inference required by the forecaster to ensure a fair comparison with baselines.

2 The paper uses S1 as a baseline method; however, this approach is relatively early. It would strengthen the evaluation if the authors could include comparisons with more recent methods.

**Limitations:**

yes

**Strengths And Weaknesses:**

### Strengths
- The proposed method and its modeling framework are novel and provide a new perspective on efficient reasoning and inference in large language models. Moreover, the concrete implementation derived from this theoretical framework demonstrates improved performance over several baselines in the math domain.

- Based on the proposed forecasting module, the paper further explores multiple downstream applications, including test-time scaling and model selection. These derived methods show promising performance and highlight the practical applicability of the proposed approach. This significantly strengthens the impact of the work.

### Weaknesses

- In the blue-highlighted part of the introduction, the paper presents a formulation of the trade-off between query difficulty and user-specific constraints. While the formulation is intuitive, the paper does not provide sufficient justification or discussion regarding its correctness or general applicability.

- In the experimental section, the forecaster is trained on the DeepScaleR-Preview dataset, which belongs to the math domain, and the evaluation is also conducted primarily on math benchmarks. As a result, the paper lacks analysis on domain mismatch or out-of-distribution scenarios between the training data of the forecaster and real-world inference tasks. This is an important concern in practice, since training data rarely covers the full spectrum of user queries. A quantitative evaluation of such scenarios would strengthen the paper.

- The descriptions of some baselines are not sufficiently clear. For example, in Figure 4, the baseline “Best forecasted of k” is not clearly described in terms of its setup and methodology. Similarly, the differences between Re-FORC and Re-FORC Pandora could be more explicitly clarified. Although these methods are discussed elsewhere in the paper, briefly explaining them in the figure caption would improve readability.

- Most experimental results are presented only as line plots. While line plots are effective at illustrating trends, they do not clearly show the exact numerical results. It would be helpful to include tables reporting representative results. In addition, when describing improvements, the paper does not clearly specify whether the reported gains are absolute or relative improvements, which further introduces ambiguity.

---

> ### Author Rebuttal · Authors · 2026-03-31
>
> We thank Reviewer C6Rw for their constructive feedback and address each point below, leading with our strongest new results.
>
> ---
>
>
> **Comment:** "In the experimental section,... of such scenarios would strengthen the paper."
>
> **Response:** We conducted experiments on coding tasks using Claude 4.5 models, representing a different domain, model family, and access paradigm, and found strong cross-domain generalization.
>
> We further ran model-routing experiments on 354 coding problems with Claude 4.5 Haiku, Sonnet, and Opus. Using an external Qwen3-4B forecaster trained on about 10k coding samples and given only the query and partial reasoning trace, **Re-FORC** matches *Sonnet-level accuracy* at *\$0.017* and *Opus-level accuracy* at *\$0.033*, outperforming both *Prompting* and **Retrieval** baselines. This is *28\% cheaper than Opus alone* and *17\% cheaper than the best non-learned baseline*, showing that Re-FORC transfers beyond mathematics to code generation and across model families, including *Qwen3-to-Claude 4.5* reward prediction. For Prompting, the forecaster produces zero-shot forecast estimates; for Retrieval, it uses a training-data datastore.
>
> More details (tables) are provided in our response to **Reviewer 55XH**.
>
> ---
>
>
> **Comment:** "In the blue-highlighted part …. correctness or general applicability."
>
> **Response:** The objective $J = E[R^* - \lambda \cdot T_{\text{total}}]$ follows the **standard Pandora's box formulation (Weitzman, 1979)**, where the goal is to maximize the expected reward minus linear search cost. In our setting we want the best answer found ($E[R^*]$) and pay proportionally for compute used ($\lambda \cdot T_{\text{total}}$). The Gittins index policy we use is an optimal policy for this objective when alternatives are independent (Weitzman, 1979), as in our model routing. Following Achille et al. (2025), we treat lambda as a user-controllable parameter rather than fixing it, since the value of computation varies by task and user preference. We show the application of our method early stopping, routing and test-time scaling.
>
> ---
>
> **Comment:** "The paper uses S1 .. comparisons with more recent methods."
>
> **Response:** We compared against DeepConf (Fu et al., 2025), a recent confidence-based method, and show that Re-FORC significantly outperforms it across all settings.
>
> For early stopping, Re-FORC achieves consistently higher accuracy than DeepConf at every token budget across all three model sizes, results can be found here – https://anonymous.4open.science/r/ICMLRebuttal-F8A7/Early_Stopping.pdf
>
> Re-FORC outperforms DeepConf by **+4.5 to +8.5 percentage points** at matched token budgets. Re-FORC provides a smooth, continuous trade-off controlled by lambda.
>
> For test-time scaling (TTS) on competition math (AMC, AIME 2024, and AIME 2025) at a budget of roughly 56k tokens, Re-FORC achieves the best *non-oracle* performance across all model sizes, reaching 59.1% (1.7B), 65.8% (4B), and 65.5% (8B). It outperforms majority voting and all *DeepConf* variants, exceeding the best DeepConf baseline by **8.7 to 14.3 percentage points** at equal token budgets. Across our full baseline suite—including majority voting, average-of-N sampling, fixed token limits, and Pass@k with oracle access—Re-FORC consistently delivers the strongest performance among all non-oracle methods.
>
> More details in our response to **Reviewer 55XH** (fig: https://anonymous.4open.science/r/ICMLRebuttal-F8A7/Early_Stopping.pdf and tables)
>
> ---
>
> **Comment:** "When reporting compute or token usage, … with baselines."
>
> **Response:**  Yes, the overhead is negligible. The forecaster is just a single self-attention pooling layer plus a linear head—about 1/28 of a transformer layer for Qwen3 1.7B and 1/37 for Qwen3 4B/8B. Since the base model’s activations are already available at inference time, the forecaster adds only a lightweight readout that predicts reward curves for all future horizons in a single forward pass. In the coding setup, the external forecaster adds only one prefill pass through Qwen3 4B over the query and partial Claude 4.5 reasoning trace, which is far cheaper than Claude’s autoregressive reasoning. We will quantify this overhead explicitly in the revised manuscript.
>
> ---
>
> **Comment:** "The descriptions of … caption would improve readability … ambiguity"
>
> **Response:** We thank the reviewer for this suggestion, we will incorporate it in our revision. "Best forecasted of k" generates k independent traces and selects the one with the highest forecaster-predicted reward (Equation 5) with no adaptive stopping or Gittins index. "Re-FORC" without Pandora refers to one-shot model selection from Equation 9, which selects the best model and thinking length at the start of inference. "Re-FORC Pandora" is the full Pandora's box greedy search from Section 4.2 that allows switching between models and trajectories during inference.

---

> > ### Author Rebuttal · Reviewer_C6Rw · 2026-04-03
> >
> > Most of my concerns are solved. I will raise my score.

---

### Official Review · Reviewer_X2WE · 2026-03-11

**Soundness:** 2
**Presentation:** 3
**Significance:** 3
**Originality:** 3
**Overall Recommendation:** 4
**Confidence:** 3

**Summary:**

This paper presents an adaptive reward prediction framework named Re-FORC. By predicting the expected reward from generating additional reasoning tokens, it optimizes the trade off between computational cost and accuracy in Chain of Thoughts reasoning for LLMs. Implemented as a lightweight adapter trained on top of a frozen inference model, the framework enables early stopping, model selection, reasoning length control, and test time scaling. Experimental results on multiple mathematical reasoning datasets demonstrate its superior performance.

**Compliance With Llm Reviewing Policy:**

Affirmed.

**Key Questions For Authors:**

1.	What is the specific training cost of Re-FORC? What is the size of the dataset required?
2.	For different base inference models, does the adapter need to be retrained?

**Limitations:**

yes

**Strengths And Weaknesses:**

Strengths:
1.	Achieving the desired objective with minimal tokens under a given query is an interesting problem.
2.	The lightweight design requires no training or architectural modifications to the base inference model, enabling fast integration into existing inference pipelines.
3.	Users can dynamically control λ at inference time to reconfigure the trade-off between accuracy and cost.
4.	The proposed method effectively mitigates the computational waste commonly observed in vanilla chain-of-thought prompting, thereby reducing the cost of model inference.

Weakness:
1.	The benchmark dataset is limited; the evaluation is only conducted on mathematical reasoning datasets, lacking experiments on cognitive reasoning tasks and code tasks.
2.	The base model is only tested on Qwen3 models of different scales; it is recommended to add experiments on other inference models.
3.	Lacking of discussion on the choice of the Beta distribution for modeling the predictor's output.

---

> ### Author Rebuttal · Authors · 2026-03-31
>
> We thank Reviewer X2WE for the constructive feedback and address each concern below.
>
> ---
>
> **Comment:** "The benchmark dataset … on cognitive reasoning tasks and code tasks."
>
> **Response:**  We conducted additional experiments on coding tasks and found that Re-FORC generalizes beyond mathematics to code generation. On the Codeforces test set, it matches *Sonnet-level accuracy at lower cost* and achieves *Opus-level accuracy with 28\% lower cost than Opus alone* and *17\% lower cost than the best non-learned baseline*, outperforming both Prompting and Retrieval.
>
> Further details are provided in our response to **Reviewer 55XH**.
>
> ---
>
> **Comment:** "The base model is only tested on Qwen3 models of different scales; it is recommended to add experiments on other inference models."
>
> **Response:**
> We conducted coding experiments with the *Claude 4.5 family* (Haiku, Sonnet, Opus), demonstrating *cross-model-family generalization*. These are **black-box API models**, so the forecaster has no access to internal activations or logits and operates only on the query and partial reasoning trace using a Qwen3-4B backbone. Despite this, it successfully predicts which Claude model will solve a given problem, suggesting that it captures general features of reasoning progress rather than model-specific internals. By contrast, the internal forecaster that reads hidden activations must be retrained for each model, though this remains lightweight since it consists of only a single attention layer and a linear head on top of a frozen base model. We will add these experiments in the revised version.
>
> ---
>
> **Comment:** "Lacking of discussion on the choice of the Beta distribution for modeling the predictor's output."
>
> **Response:**
> We use a Beta distribution for four reasons: **(1) bounded support**, since the predictor outputs probabilities in [0,1]; **(2) flexibility**, since $\text{Beta}(\alpha,\beta)$ can capture a wide range of confidence patterns with just two parameters; **(3) uncertainty quantification**, since it provides both a mean prediction and a variance estimate; and **(4) numerical stability**, since its likelihood is easy to optimize and softplus ensures $(\alpha,\beta>0)$ during training. Overall, the Beta distribution is a simple and well-suited choice for modeling probabilistic correctness predictions. We will include this discussion in the revised manuscript.
>
> ---
>
> **Comment:** "What is the specific training cost of Re-FORC? What is the size of the dataset required?"
>
> **Response:**
>
> The training cost is modest relative to training the base model.
>
> For the math forecaster, we train on DeepScaleR-Preview by generating one full reasoning trajectory per problem and extracting partial traces at the 17 grid points T={0,512,..,8192}. At each point, we generate N=8 answer completions to estimate expected reward, so the dominant cost is at most 136 answer generations per problem, per model. Our trajectory reuse strategy (Section 3.3) shares the same reasoning trace across all grid points, avoiding redundant computation.
>
> For the coding forecaster, we use only about 10k coding samples, roughly 4× fewer than in math, yet still obtain strong performance.
>
> Training is also lightweight because the forecaster is just a single attention layer plus a linear head on top of a frozen base model, requiring only activation extraction rather than full-model optimization. Overall, the total overhead of data generation and forecaster training is orders of magnitude smaller than pretraining or fine-tuning the base reasoning model. We will report exact GPU hours in the revised manuscript.
>
> ---
>
> **Comment:** "For different base inference models, does the adapter need to be retrained?"
>
> **Response:** Yes, the forecaster requires retraining for different base models to calibrate its predictions to their specific output distributions. Our coding experiments demonstrate this: the Qwen3 4B forecaster was trained on approximately 10k coding samples from Claude 4.5 models to predict their rewards. However, retraining is lightweight since the forecaster is just a single attention layer plus a linear head on top of a frozen base model. We hypothesize that finetuning a forecaster already trained on other models should be more sample efficient than training from scratch, though we leave empirical validation of this to future work.
>
> ---

---

> > ### Author Rebuttal · Reviewer_X2WE · 2026-04-03
> >
> > The authors state that experiments were conducted on the “Claude 4.5 family,” but where are the corresponding results? Why were other open-source models not included?

---

> > > ### Author Response · Authors · 2026-04-05
> > >
> > > We thank Reviewer X2WE for the follow-up question and clarify below.
> > >
> > > **Claude 4.5 results.** The corresponding results are provided in our response to Reviewer 55XH. For convenience, we reproduce the key table here:
> > >
> > > | Target Accuracy | Re-FORC | Prompting | Retrieval | Standalone |
> > > |---|---|---|---|---|
> > > | Sonnet (75.9%) | $0.017 | $0.019 | $0.020 | $0.021 |
> > > | Opus (83.5%) | $0.033 | $0.040 | $0.042 | $0.046 |
> > >
> > > Re-FORC achieves Opus-level accuracy (83.5%) at **28% lower cost** than running Opus alone and **17% cheaper** than the best non-learned baseline. The external forecaster uses a Qwen3-4B backbone trained on ~10k CodeForces samples and operates in a fully black-box setting, with no access to internal activations or logits.
> > >
> > > **Why Claude 4.5 rather than additional open-source models.** We found that the Qwen3 models used in our math experiments (1.7B, 4B, 8B) perform poorly on agentic coding tasks, which can span 32k+ tokens of reasoning, well beyond the effective range of these small models. Their base accuracy was too low to meaningfully evaluate routing and early stopping. We therefore chose the Claude 4.5 family (Haiku, Sonnet, Opus), which provides a natural hierarchy of cost and capability suitable for coding. Importantly, this choice also demonstrates that Re-FORC generalizes to a completely different model family in a strictly harder, black-box API setting where logits are unavailable. The fact that a Qwen3-4B forecaster can successfully predict rewards and route across Claude 4.5 models using only the query and partial reasoning trace provides strong evidence of cross-model-family generalization. That said, would the reviewer appreciate additional experiments with a strong open-source model (e.g., Qwen/Qwen3-Coder-Next) for the camera-ready? This would require significant compute investment that we cannot complete before the rebuttal deadline, but we can attempt it for the final version if the reviewer considers it important for their assessment.
> > >
> > > Finally, while the above addresses your specific question, we want to additionally share results from experiments requested by Reviewer 55XH on out-of-distribution generalization. To test whether a forecaster trained on one domain can generalize to a genuinely unseen domain, we ran an OOD test-time-scaling experiment on **MMLU-Pro** using a forecaster trained only on math data (without any MMLU-Pro training). We have the following:
> > >
> > > | Model | Re-FORC | DeepConf Online 10% | DeepConf Offline Tail Conf | Majority Vote |
> > > |---|---|---|---|---|
> > > | 1.7B | 33.7% | 33.4% | **35.3%** | 33.5% |
> > > | 4B | **39.4%** | 37.5% | 39.1% | 39.0% |
> > > | 8B | **39.9%** | 38.8% | 39.3% | 38.5% |
> > >
> > > Re-FORC **outperforms all baselines for the 4B and 8B models** (+0.3pp and +0.6pp respectively), while DeepConf Tail Conf is stronger for 1.7B. This is consistent with our paper's finding that forecast quality scales with model capacity—"forecast quality improves as chain-of-thought tokens increase... this effect is amplified in larger models"—and the 1.7B model's "limited capability" noted in our analysis. The weaker OOD performance on MMLU-Pro for 1.7B aligns with this pattern where smaller models struggle more on out-of-domain tasks. This does not prove the forecaster is fully domain-agnostic, but shows the forecasting signal can transfer usefully beyond the training domain for sufficiently capable models.

---

### Official Review · Reviewer_DBNE · 2026-03-13

**Soundness:** 3
**Presentation:** 3
**Significance:** 3
**Originality:** 3
**Overall Recommendation:** 4
**Confidence:** 4

**Summary:**

This paper proposes Re-FORC, an adaptive inference framework based on future reward prediction, to improve computational allocation efficiency in chain-of-thought reasoning for large language models. The method augments existing reasoning models with a lightweight forecaster that predicts the expected reward achievable by generating varying numbers of thinking tokens from the current inference state. The results demonstrate that, without training the base reasoning model, this approach reduces computational overhead while maintaining accuracy, achieves higher accuracy under the same compute budget, or significantly lowers computational costs at equivalent accuracy levels.

**Compliance With Llm Reviewing Policy:**

Affirmed.

**Final Justification:**

I appreciate the authors' additional experiments, my concerns have been addressed. Since I have already given a positive score of 4, I will maintain my current rating.

**Key Questions For Authors:**

See weaknesses

**Limitations:**

yes

**Strengths And Weaknesses:**

**Strength**

- 1. The method is well-conceived with architectural consistency, extending beyond early stopping to explicitly model the future reward–compute trade-off.

- 2. The inference-time control requires minimal hyperparameter tuning, and critically, demands no modification or retraining of the base reasoning model—only the attachment of a lightweight forecasting adapter is needed.

**Weakness**

- 1. While the paper demonstrates favorable performance-efficiency trade-offs across different budget settings, the forecaster requires training. Therefore, comparisons against existing training-based adaptive reasoning methods are necessary. Additionally, lightweight entropy-based inference time adaptive reasoning methods should also be included as baselines.

- 2. The experiments are primarily confined to mathematical reasoning. For practical deployment, it remains unclear whether a single forecaster can be shared across domains or if domain-specific training is required—this represents a significant gap given the broader use cases of reasoning models. Furthermore, the compatibility of this forecaster with mainstream inference frameworks (e.g., vLLM) warrants investigation.

- 3. The paper lacks clarity regarding the volume of training data required to achieve the claimed performance for each model scale. This is critical for assessing scalability to larger models (≥30B parameters) with more data that are more commonly used for inference.

---

> ### Author Rebuttal · Authors · 2026-03-31
>
> We thank Reviewer DBNE for their constructive feedback and below we address each concern with new experimental evidence.
>
> ---
>
> **Comment:** "While the paper .. lightweight entropy-based inference time adaptive reasoning … baselines."
>
> **Response:**  We agree that entropy-based adaptive reasoning methods are important baselines. Accordingly, we added experiments against the baseline **DeepConf** (Fu et al., 2025), which uses entropy-based confidence for enhanced reasoning. Across all model sizes and token budgets, **Re-FORC consistently outperforms DeepConf and all other baselines**. For early stopping, Re-FORC achieves higher accuracy than DeepConf at matched budgets; for Test-Time-Scaling, it attains the best **non-oracle** performance among all baselines.
>
> We provide detailed results (fig: https://anonymous.4open.science/r/ICMLRebuttal-F8A7/Early_Stopping.pdf and tables) in our response to **Reviewer 55XH**.
>
> We also note that *Re-FORC requires training only a lightweight forecaster*, rather than fine-tuning an LLM for next-token prediction or reinforcement learning. As such, comparison to training-based methods is not fully like-for-like. In addition, our method applies naturally to **black-box models**, such as the **Claude 4.5 family** (Haiku, Sonnet, Opus), where **logits are unavailable**.
>
> ---
>
> **Comment:** "The experiments …. mathematical reasoning …. (e.g., vLLM) warrants investigation."
>
> **Response:** We address all three sub-concerns with new experiments and discussion.
>
> Regarding cross-domain generalization, we conducted additional experiments on coding tasks and found that **Re-FORC generalizes beyond mathematics to code generation**. On the Codeforces test set, Re-FORC matches **Sonnet-level accuracy at lower cost** and achieves **Opus-level accuracy at 28\% lower cost than Opus alone** and **17\% lower cost than the best non-learned baseline**, outperforming both Prompting and Retrieval. For*Prompting, the forecaster produces zero-shot forecast estimates; for Retrieval, it uses a training-data datastore. This shows the generalizability of Re-FORC to black-box LLM settings, such as the API based LLMs.
>
> Additional details / tables are provided in our response to **Reviewer 55XH**.
>
> Regarding single forecaster across domains, our current approach trains lightweight domain-specific forecasters (one for math, one for coding) given the modest training cost. The ability of a single forecaster to serve multiple domains without domain-specific training is interesting future work. The external forecaster architecture, which processes only text and does not depend on domain-specific internal representations, is a promising candidate for this.
>
> Regarding vLLM compatibility, Re-FORC forecaster is architecturally compatible with standard inference frameworks including vLLM. For internal forecaster mode, the adapter reads penultimate-layer activations already computed during autoregressive generation. No changes to the generation loop are needed, only an additional lightweight forward pass through the adapter at decision points (every $\Delta=512$ tokens). For external forecaster mode, the forecaster runs as a completely separate model communicating only via text, making it trivially compatible with any serving framework, including vLLM, TGI, or proprietary APIs.
>
>
> **Comment:** "The paper lacks clarity … models (>=30B parameters) with … used for inference."
>
> **Response:**  We discuss this in Section 5.1, but will elaborate further:
> For the math forecaster, we train on DeepScaleR-Preview, comprising roughly 40k problems. For each problem and model scale (1.7B, 4B, 8B), we generate one reasoning trajectory and extract partial traces at the 17 grid points T={0,512,...,8192}. At each point, we generate N=8 answer completions to estimate expected reward, making the dominant cost at most 136 answer generations per problem, per model. Section 3.3’s trajectory reuse strategy avoids redundant reasoning by sharing the same trace across all grid points. For the coding forecaster, we train on about 10k CodeForces samples per model. The forecaster architecture is identical across scales: a single self-attention pooling layer followed by a linear head.
> Re-FORC supports scalability to 30B+ models – first, the main training cost comes from data generation rather than adapter training, since the adapter is lightweight and trained on a frozen base model. Second, the external forecaster decouples scale: a small Qwen3 4B model can predict rewards for much larger models using only their text outputs. Our coding experiments already show this, with a 4B forecaster successfully routing across Claude 4.5 Haiku, Sonnet, and Opus. This setup requires no per-scale adapter retraining and could extend naturally to 30B+ models without changing the forecaster architecture, which we leave as future work.

---

### Decision · Program_Chairs · 2026-04-30

**Decision:**

Accept (regular)

**Comment:**

This paper proposes **Re-FORC**, a lightweight reward-forecasting framework for adaptive compute allocation in reasoning models. By predicting expected future reward as a function of additional thinking tokens, the method supports several inference-time decisions, including early stopping, model selection, and test-time scaling. The problem is timely and important, and the paper presents a coherent approach that connects reward forecasting to sequential compute-allocation policies in a practically useful way.

The paper has several strengths. Reviewers generally found the core idea interesting, and the broader framework enables multiple downstream use cases beyond a single early-exit setting. The forecasting module is lightweight, requires no retraining of the base reasoning model, and can be controlled at inference time through a simple trade-off parameter. The rebuttal also materially strengthened the empirical case by adding comparisons to stronger baselines, clarifying training cost and forecaster overhead, and expanding the evaluation beyond the original math-only setting.

At the same time, I do not think the current evidence fully supports the strongest version of the paper’s framing. In particular, the most convincing interpretation of the results is that Re-FORC is a **lightweight and empirically effective adaptive-compute method with promising initial OOD results**, rather than a fully established general forecasting framework that transfers broadly across domains and model families without retraining. The new math-trained forecaster result on MMLU-Pro is useful and strengthens the paper meaningfully, but it remains limited in scope, and the broader domain-agnostic claim should be stated more carefully. Similarly, while the coding experiments are valuable, they mostly demonstrate applicability with additional domain-specific forecaster training rather than complete cross-domain generality.

Overall, I find the paper solid and above the bar, but with limitations that should temper the framing. The rebuttal addressed most of the major concerns in a substantive way, and the remaining issues are mainly about scope calibration rather than fundamental soundness. I therefore recommend **weak accept**. In the final version, the paper should present the contribution primarily as a strong adaptive-compute method with promising but still limited out-of-domain evidence, and avoid overstating the extent to which a single forecaster is already shown to generalize across arbitrary unseen domains.